# GEX: A flexible method for approximating influence via Geometric Ensemble

**Sung-Yub Kim**
Graduate School of AI, KAIST
sungyub.kim@kaist.ac.kr

**Kyungsu Kim**[†]
Massachusetts General Hospital and Harvard Medical School
kskim.doc@gmail.com

**Eunho Yang**[†]
Graduate School of AI, KAIST and AITRICS
eunhoy@kaist.ac.kr

## Abstract

Through a deeper understanding of predictions of neural networks, Influence Function (IF) has been applied to various tasks such as detecting and relabeling mislabeled samples, dataset pruning, and separation of data sources in practice. However, we found standard approximations of IF suffer from performance degradation due to oversimplified influence distributions caused by their bilinear approximation, suppressing the expressive power of samples with a relatively strong influence. To address this issue, we propose a new interpretation of existing IF approximations as an average relationship between two linearized losses over parameters sampled from the Laplace approximation (LA). In doing so, we highlight two significant limitations of current IF approximations: the linearity of gradients and the singularity of Hessian. Accordingly, by improving each point, we introduce a new IF approximation method with the following features: i) the removal of linearization to alleviate the bilinear constraint and ii) the utilization of Geometric Ensemble (GE) tailored for non-linear losses. Empirically, our approach outperforms existing IF approximations for downstream tasks with lighter computation, thereby providing new feasibility of low-complexity/nonlinear-based IF design.

## 1 Introduction

In the last decade, neural networks (NNs) have made tremendous advances in various application areas [49, 24, 47]. To make reasonable predictions with NN-based systems, models must be able to explain their predictions. For example, those who doubt the model's prediction can gain insight and foresight by referencing the explanation of the model. Moreover, mission-critical areas like finance and medicine require a high degree of explainability to ensure that the predictions are not biased [30]. Understanding the mechanism of predictions also allows researchers and engineers to improve prediction quality, ensuring that NNs are performing as intended [32].

To this end, Influence Function (IF) was proposed to explain predictions of pre-trained NNs through training data [25]. Intuitively, IF measures how the leave-one-out (LOO) retraining of a training sample changes the loss of each sample. Therefore, the sign of influence determines whether the training sample is beneficial to others, and the scale of influence measures its impact. Specifically, self-influence, the increase in loss when a sample is excluded, was used to measure how much the sample is memorized [48, 15]: When LOO training is performed on a memorized training sample, its loss will increase substantially since matching its (corrupted) label will be difficult. Therefore, self-influence is used to detect mislabeled samples [48, 52] where memorization occurs. Furthermore,

37th Conference on Neural Information Processing Systems (NeurIPS 2023).

[†] Correspondence to

recent works successfully applied IF to various downstream tasks, including dataset pruning [58] and data resampling [59, 63].

Despite its broad applicability, we found that IF and its approximations [25, 48, 52] suffer from oversimplified self-influence distributions due to their bilinear form[1]. Although these approximations are introduced to avoid prohibitive retraining costs of IF, they impose a structural constraint that self-influence becomes quadratic to gradients of pre-trained NNs. Due to this constraint, self-influence follows an unimodal distribution, as gradients of pre-trained NNs typically follow a zero-centered Stable distribution [7, 56]. Unfortunately, unimodal distributions are too restrictive for representing self-influence in real-world datasets containing mislabeled samples. While self-influence distributions estimated by LOO retraining may become bimodal depending on the proportion of (high self-influential) mislabeled samples, unimodal distributions cannot handle this case.

To resolve this problem, we propose a non-linear IF approximation via Geometric Ensemble (GE; [16]). Our method is motivated by a novel connection between IF approximation and linearized Laplace approximation (LA; [38]) that we discovered: IF approximations can be translated to an averaged relationship between two linearized losses over parameters sampled from LA. As linearized losses in this connection cause bilinear forms of IF approximations, we consider an IF approximation without linearization. However, we then identify an additional issue of this approximation due to the singularity of Hessian and its solutions (e.g., damping and truncation). To mitigate this issue, we propose a novel approach using GE to manage the relationship of non-linear losses more effectively. As a result, our approach, **G**eometric **E**nsemble for sample e**X**planation (GEX), accurately represents the multimodal nature of LOO retraining, leading to improved performance in downstream tasks across various scenarios. Furthermore, $\mathcal{I}_{\texttt{GEX}}$ is easy to estimate as it does not require Jacobian-vector products (JVPs) for batch estimation or sub-curvature approximations like LA.

We summarize our contributions as follows:

- We identify a distributional bias in commonly used IF approximations. We demonstrate how this bias results in oversimplified distributions for self-influences.

- We provide a novel connection between IF approximations and LA. By identifying an inherent issue of LA, we provide a non-linear IF approximation via GE, named $\mathcal{I}_{\texttt{GEX}}$. Due to its non-linear nature, $\mathcal{I}_{\texttt{GEX}}$ can express various influence distributions depending on the characteristic of the datasets.

- We verify that $\mathcal{I}_{\texttt{GEX}}$ outperforms standard IF approximations in downstream tasks, including noisy label detection, relabeling, dataset pruning, and data source separation. We also show that $\mathcal{I}_{\texttt{GEX}}$ is competitive with well-known baselines of downstream tasks with lighter computation.[2]

## 2 Background

Consider an independently and identically (i.i.d.) sampled training dataset $S := \{z_n : (x_n, y_n)\}_{n=1}^N$ where $x_n \in \mathbb{R}^D$ is an input vector of $n$-th sample and $y_n \in \mathbb{R}^K$ is its label. To model the relation between inputs and outputs of training samples, we consider a neural network (NN) $f : \mathbb{R}^D \times \mathbb{R}^P \to \mathbb{R}^K$ which maps input $x \in \mathbb{R}^D$ and network parameter $\theta \in \mathbb{R}^P$ to a prediction $\hat{y}_n := f(x_n, \theta) \in \mathbb{R}^K$. Empirical risk minimization (ERM) solves the following optimization problem

$$\theta^* := \underset{\theta \in \mathbb{R}^P}{\operatorname{argmin}} \, L(S, \theta)$$

where $L(S, \theta) := \sum_{n=1}^N \ell(z_n, \theta)/N$ for a sample-wise loss $\ell(z_n, \theta)$ (e.g., cross-entropy between $\hat{y}_n$ and $y_n$). In general, $\theta^*$ is trained using a stochastic optimization algorithm (e.g., stochastic gradient descent (SGD) with momentum). For the reader's convenience, we provide the notation table in Appendix A for terms used in the paper.

The Leave-One-Out (LOO) retraining effect of $z \in S$ on another instance $z' \in \mathbb{R}^D$ is defined as the difference of sample-loss $z'$ between the $\theta^*$ and the retrained point $\theta_z^*$ without $z$ [9]:

$$\mathcal{I}_{\texttt{LOO}}(z, z') := \ell(z', \theta_z^*) - \ell(z', \theta^*) \tag{1}$$

---

[1]Here, the bilinearity refers to the IF and its approximations being bilinear with respect to the sample-wise gradients. Consequently, the self-influence based on these bilinear metrics is quadratic for sample-wise gradients.

[2]Code is available at https://github.com/sungyubkim/gex.

where $\theta_z^* := \arg\min_{\theta \in \mathbb{R}^P} L(S, \theta) - \ell(z, \theta)/N$. Since retraining for every pair $(z, z')$ is computationally intractable where we have a huge number of data points as in practice, Koh and Liang [25] proposed an efficient approximation of $\mathcal{I}_{\texttt{LOO}}$, named Influence Function (IF):

$$\mathcal{I}(z, z') := g_{z'}^\top H^{-1} g_z \tag{2}$$

where $g_z := \nabla_\theta \ell(z, \theta^*) \in \mathbb{R}^P$ and $H := \nabla_\theta^2 L(S, \theta^*) \in \mathbb{R}^{P \times P}$ by assuming the strictly convexity of $\ell$ (i.e., $H$ is positive definite). Here, $\mathcal{I}$ can be understood as a two-step approximation of $\mathcal{I}_{\texttt{LOO}}$

$$\mathcal{I}_{\texttt{LOO}}(z, z') \approx \ell_{\theta^*}^{\lin}(z', \theta_z^*) - \ell^{\lin}(z', \theta^*) = g_{z'}^\top(\theta_z^* - \theta^*) \tag{3}$$

$$\approx g_{z'}^\top H^{-1} g_z = \mathcal{I}(z, z') \tag{4}$$

where $\ell_{\theta^*}^{\lin}(z, \psi) := \ell(z, \theta^*) + g_z^\top(\psi - \theta^*)$ and $\psi \in \mathbb{R}^P$ is an arbitrary vector in the parameter space $\mathbb{R}^P$. Here, we use the superscript $\lin$ to indicate linearization. Note that (3) applies linearization to the sample-loss $z'$ and (4) approximates parameter difference as a Newton ascent term (i.e., $\theta_z^* \approx \theta^* + H^{-1} g_z$).

While the computation of $\mathcal{I}$ is cheaper than $\mathcal{I}_{\texttt{LOO}}$, it is still intractable to modern NNs (e.g., ResNet [20] and Transformer [62]) because of the prohibitively large Hessian. To alleviate this problem, two additional approximations are commonly used: stochastic approximation methods such as $\mathcal{I}_{\texttt{LiSSA}}$ [1], and sub-curvature approximations such as $\mathcal{I}_{\texttt{Last-Layer}}$, which limits the Hessian computation only to the last-layer of NNs [10]. However, both methods also have their own problems: $\mathcal{I}_{\texttt{LiSSA}}$ takes high time complexity since the inverse Hessian-vector product (IHVP) for each training sample needs to be computed separately, as shown in Schioppa et al. [52]. On the other hand, $\mathcal{I}_{\texttt{Last-Layer}}$ may cause inaccurate IF approximations, as shown in Feldman and Zhang [15].

As another alternative, Pruthi et al. [48] recently proposed to exploit intermediate checkpoints during pre-training, named $\mathcal{I}_{\texttt{TracIn}}$:

$$\mathcal{I}_{\texttt{TracIn}}(z, z') := \frac{1}{C} \sum_{c=1}^C g_{z'}^{c\top} g_z^c \tag{5}$$

where $g_z^c := \nabla_\theta \ell(z, \theta^c)$ for checkpoints $\theta^c$ ($c = 1, \ldots, C$) sampled from the pre-training trajectory of $\theta^*$. Here, it further simplifies the computation by assuming the expensive $H^{-1}$ in (2) is an identity matrix. Instead, the performance of $\mathcal{I}_{\texttt{TracIn}}$ is replenished by averaging over several intermediate checkpoints, which capture various local geometries of loss landscapes. In addition, Pruthi et al. [48] enhanced the efficiency of $\mathcal{I}_{\texttt{TracIn}}$ using **random projection**, named $\mathcal{I}_{\texttt{TracInRP}}$,

$$\mathcal{I}_{\texttt{TracInRP}}(z, z') := \frac{1}{C} \sum_{c=1}^C g_{z'}^{c\top} Q_R Q_R^\top g_z^c \tag{6}$$

where $Q_R \in \mathbb{R}^{P \times R}$ is a random projection matrix whose components are i.i.d. sampled from $\mathcal{N}(0, 1/R)$ for $R \ll P$. Note that $\mathcal{I}_{\texttt{TracInRP}}$ is an unbiased estimator of $\mathcal{I}_{\texttt{TracIn}}$ as $\mathbb{E}[Q_R Q_R^\top] = \mathbf{I}_P$ where $\mathbf{I}_P$ is the identity matrix of dimension $P \times P$. However, $\mathcal{I}_{\texttt{TracIn}}$ and $\mathcal{I}_{\texttt{TracInRP}}$ cannot be applied to checkpoints of the open-source community, such as TorchHub [45] and Huggingface models [65] since they only provide the final checkpoints without any intermediate results during pre-training.

On the other hand, Schioppa et al. [52] proposed to approximate IF in a purely post-hoc manner as

$$\mathcal{I}_{\texttt{Arnoldi}}(z, z') := g_{z'}^\top U_R \Lambda_R^{-1} U_R^\top g_z \tag{7}$$

where $\Lambda_R \in \mathbb{R}^{R \times R}$ is a diagonal matrix whose elements are top-$R$ eigenvalues of $H$ and the columns of $U_R \in \mathbb{R}^{P \times R}$ are corresponding eigenvectors. They use Arnoldi iteration [3] to estimate $\Lambda_R$ and $U_R$. Contrary to $\mathcal{I}_{\texttt{TracInRP}}$ with the random projection, $\mathcal{I}_{\texttt{Arnoldi}}$ projects $g_z, g_{z'}$ to the top-$R$ eigenspace of the Hessian. Due to this difference, Schioppa et al. [52] argued that $\mathcal{I}_{\texttt{Arnoldi}}$ perform comparably to $\mathcal{I}_{\texttt{TracInRP}}$ with less $R$.

Finally, it is important to note that $\mathcal{I}_{\texttt{TracInRP}}$ and $\mathcal{I}_{\texttt{Arnoldi}}$ can perform **batch computations** using Jacobian-vector products (JVPs). To be more precise, computing $\mathcal{I}$ and $\mathcal{I}_{\texttt{TracIn}}$ for multiple samples requires sample-wise gradient computation for each $g_z, g_z^c$, which is difficult to parallelize because of heavy memory complexity. In contrast, $\mathcal{I}_{\texttt{TracInRP}}$ and $\mathcal{I}_{\texttt{Arnoldi}}$ can avoid this sample-wise computation by computing JVPs for a batch at once (i.e., parallelize $g_z^{c\top} Q_R, g_z^\top U_R$ for multiple $z$).

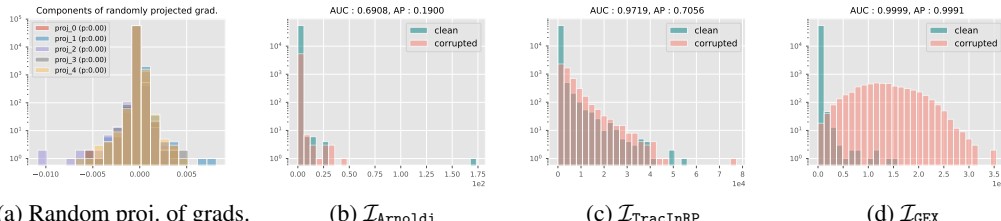

(a) Random proj. of grads.      (b) $\mathcal{I}_{\texttt{Arnoldi}}$      (c) $\mathcal{I}_{\texttt{TracInRP}}$      (d) $\mathcal{I}_{\texttt{GEX}}$

Figure 1: Since pre-trained gradients follow Gaussian distributions (Fig. 1(a)), quadratic self-influences follow unimodal distributions (See Proposition 3.1), even in noisy label settings (Figs. 1(b)-1(c)). This issue can be solved by removing linearization in the IF approximation (Fig. 1(d)).

## 3    Identifying distributional bias in bilinear influence approximations

Throughout the discussion above, IF approximations ((2), (5), (6), and (7)) are defined as **bilinear forms of gradients**. In this section, we demonstrate a side effect of this form on computing self-influence. Our key empirical observation is that **gradients of pre-trained NNs follow Gaussian distributions centered at zero**. To verify this, we train VGGNet [55] on MNIST [33] with 10% label corruption following Schioppa et al. [52]. Then, we project $\{g_z\}_{z \in S}$ onto each of 5 random vectors $\{d_i\}_{i=1}^5$ (with $d_i \in \mathbb{R}^P$), uniformly sampled on the unit sphere. If a pre-trained gradient $g_z$ follows Gaussian distribution, its projected component $g_z^\top d_i$ will also follow Gaussian distributions for all $i = 1, \ldots, 5$. We visualize histograms of projected components $g_z^\top d_i$ for each $d_i$, and the resulting p-values of the normality test [57] in Fig. 1(a). In agreement with our hypothesis, the randomly projected components follow Gaussian distributions with significantly low p-values. Moreover, one can observe that Gaussian distributions are centered at zero. This is because gradients at $\theta^*$ satisfy

$$\mathbb{E}_{z \sim S} \left[ g_z^\top d \right] = \frac{1}{N} \sum_{z \in S} g_z^\top d = \mathbf{0}_P^\top d = 0$$

by the first-order optimality condition at $\theta^*$ (i.e., $\nabla_\theta L(S, \theta^*) = \mathbf{0}_P$) where $\mathbf{0}_P$ is the zero vector of dimension $P$. Note that a similar observation on the normality of gradient was reported in the context of differential privacy in Chen et al. [7].

Now let us write the eigendecomposition of Hessian as $H = \sum_{i=1}^P \lambda_i u_i u_i^\top$ where $\lambda_1, \ldots, \lambda_P > 0$ are eigenvalues of $H$ in descending order and $u_1, \ldots, u_P \in \mathbb{R}^P$ are corresponding eigenvectors by positive definite assumption of Koh and Liang [25]. We then arrange self-influence of $\mathcal{I}$ as

$$\mathcal{I}(z, z) = g_z^\top H^{-1} g_z = \sum_{i=1}^P \frac{(g_{z,i})^2}{\lambda_i} \tag{8}$$

where $g_{z,i} := g_z^\top u_i$ are $i$-th component of $g_z$ in the eigenspace of $H$. Following the above discussion, one can assume that $g_{z,i}$ follows a Gaussian distribution. Consequently, $\mathcal{I}(z, z)$ follows a (generalized) $\chi^2$-distribution due to squared components in (8). The following proposition shows that this phenomenon can be generalized to any stable distribution [56] and positive definite matrix.

**Proposition 3.1** (Distributional bias in bilinear self-influence). *Let us assume $g_z$ follows a $P$-dimensional stable distribution (e.g., Gaussian, Cauchy, and Lévy distribution) and $M \in \mathbb{R}^{P \times P}$ is a positive (semi-)definite matrix. Then, self-influence in the form of $\mathcal{I}_M(z, z) = g_z^\top M g_z$ follows a unimodal distribution. Furthermore, if $g_z$ follows a Gaussian distribution, then the self-influence follows a generalized $\chi^2$-distribution.*

We refer to Appendix B for the proof. Self-influence distributions approximated by $\mathcal{I}_{\texttt{Arnoldi}}$ (Fig. 1(b)) and $\mathcal{I}_{\texttt{TracInRP}}$ (Fig. 1(c)) provide the empirical evidence of Proposition 3.1 in the noisy label setting in Fig. 1(a). In this setting, each mislabeled sample exhibits high self-influence values since predictions by incorrect labels are challenging to recover when removed and retrained. Therefore, mislabeled samples constitute a distinct mode with a greater self-influence than correctly labeled samples. Still, the distributional bias in Proposition 3.1 squeezes correct and mislabeled samples in a unimodal distribution. As a result, two types (clean and corrupted) of samples are indistinguishable in $\mathcal{I}_{\texttt{Arnoldi}}$ and $\mathcal{I}_{\texttt{TracInRP}}$. Proposition 3.1 shows that this observation can be generalized to a more diverse setting (e.g., heavy-tailed distributions). In Sec. 5.1, we demonstrate that this bias occurs regardless of the dataset and the architectures of NN.

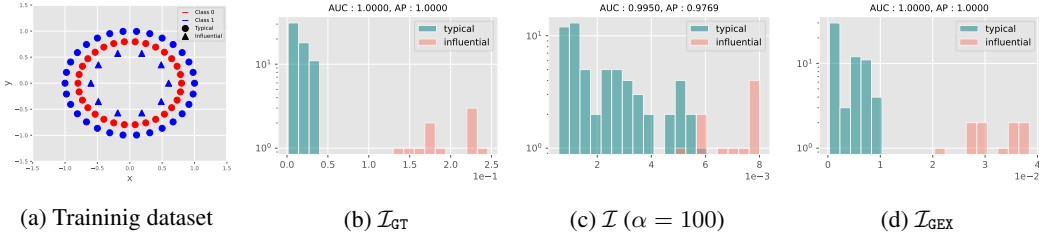

|  |  |  |  |
|:---:|:---:|:---:|:---:|
| (a) Traininig dataset | (b) $\mathcal{I}_{\text{GT}}$ | (c) $\mathcal{I}$ ($\alpha = 100$) | (d) $\mathcal{I}_{\text{GEX}}$ |

Figure 2: Fig. 2(a): The modified two-circle dataset in Sec. 4. Here, the typical samples are the two outer circle samples with relatively high density, and influential samples correspond to the inner circle, demonstrating relatively low density. The self-influence histogram estimated with $\mathcal{I}_{\text{LOO}}$, $\mathcal{I}$, and $\mathcal{I}_{\text{GEX}}$ are provided in Fig. 2(b)-2(d). Fig. 2(b): $\mathcal{I}_{\text{LOO}}$ properly separates the influential samples from the typical samples in the modified two-circle dataset. Fig. 2(c): $\mathcal{I}$ mixes typical and influential samples due to the distributional bias in Sec. 3. Fig. 2(d): $\mathcal{I}_{\text{GEX}}$ accurately represents the bimodal nature of $\mathcal{I}_{\text{LOO}}$.

## 4 Geometric Ensemble for sample eXplanation

To mitigate the distributional bias in Sec. 3, we propose a flexible IF approximation method using Geometric Ensemble (GE; [16]), named Geometric Ensemble for sample eXplanataion (GEX). Here is a summary of how GEX is developed.

$$\mathcal{I} \xrightarrow[\text{Section 4.1}]{\text{Delinearization}} \mathcal{I}_{\text{LA}} \xrightarrow[\text{Section 4.2}]{\text{LA to GE}} \mathcal{I}_{\text{GEX}}$$

In Sec. 4.1, we ensure that the influence approximation is not a bilinear form for the gradient by replacing gradients in IF with sample-loss deviations. The theoretical foundation for this step is provided by our Theorem 4.1 below, which establishes a relationship between the IF and the Laplace approximation (LA; [38]). Moving on to Sec. 4.2, we modify the parameter distribution to compute the sample-loss deviation from LA to GE. This modification is necessary because GE is based on the local geometry of the loss landscape around $\theta^*$, similar to LA, while avoiding overestimating loss deviations caused by the singularity of the Hessian.

### 4.1 GEX and its motivation

The proposed method, GEX, comprises three steps. The first step involves the collection of post-hoc checkpoints $\{\theta^m\}_{m=1}^M$ through multiple SGD updates on the training loss starting from $\theta^*$. Then, the empirical distribution of Geometric Ensemble (GE) is computed as

$$p_{\text{GE}}(\psi) := \frac{1}{M} \sum_{m=1}^M \delta_{\theta^{(m)}}(\psi). \tag{9}$$

where $\delta_{\theta^{(m)}}(\cdot)$ denotes the Dirac delta distribution at $\theta^{(m)}$. In the final step, GEX is obtained as the expectation of the product of **sample-loss deviations** from the pre-trained parameter as follows

$$\mathcal{I}_{\text{GEX}}(z, z') = \mathbb{E}_{\psi \sim p_{\text{GE}}} \left[ \Delta \ell_{\theta^*}(z, \psi) \cdot \Delta \ell_{\theta^*}(z', \psi) \right] \tag{10}$$

where $\Delta \ell_{\theta^*}(z, \psi) := \ell(z, \psi) - \ell(z, \theta^*)$ means the sample-loss deviations from $\theta^*$. We provide the pseudocode for computing $\mathcal{I}_{\text{GEX}}$ in Appendix C.

The main motivation behind $\mathcal{I}_{\text{GEX}}$ in (10) is to establish a new connection between $\mathcal{I}$ and LA. This connection is demonstrated in Theorem 4.1, which shows that $\mathcal{I}$ is an expectation of the product of **linearized sample-loss deviations** given that parameters are sampled from LA.

**Theorem 4.1** (Connection between IF and LA). *$\mathcal{I}$ in Koh and Liang [25] can be expressed as*

$$\mathcal{I}(z, z') = \mathbb{E}_{\psi \sim p_{\text{LA}}} \left[ \Delta \ell_{\theta^*}^{\text{lin}}(z, \psi) \cdot \Delta \ell_{\theta^*}^{\text{lin}}(z', \psi) \right] \tag{11}$$

*where $\Delta \ell_{\theta^*}^{\text{lin}}(z, \psi) := \ell_{\theta^*}^{\text{lin}}(z, \psi) - \ell_{\theta^*}^{\text{lin}}(z, \theta^*) = g_z^\top (\psi - \theta^*)$ and $p_{\text{LA}}$ is the Laplace approximated posterior*

$$p_{\text{LA}}(\psi) := \mathcal{N} \left( \psi | \theta^*, H^{-1} \right).$$

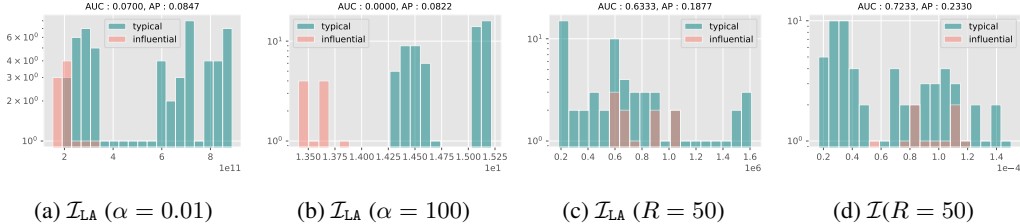

(a) $\mathcal{I}_{\text{LA}}$ ($\alpha = 0.01$)  (b) $\mathcal{I}_{\text{LA}}$ ($\alpha = 100$)  (c) $\mathcal{I}_{\text{LA}}$ ($R = 50$)  (d) $\mathcal{I}(R = 50)$

Figure 3: Damping trick for singular Hessian causes severe overestimation of self-influence (Fig. 3(a)). This issue cannot be addressed with a large damping coefficient (Fig. 3(b)). Although truncating small eigenvalues can reduce the overestimation of self-influence (Fig. 3(c)), it introduces another error that even occurs in $\mathcal{I}$ (Fig. 3(d)).

The LA was proposed to approximate the posterior distribution with a Gaussian distribution. Recently, it has gained significant attention due to its simplicity and reliable calibration performance [50, 11]. Intuitively, LA is equivalent to the second-order Taylor approximation of log-posterior at $\theta^*$ with Gaussian prior defined as $p(\psi) := \mathcal{N}(\psi|\mathbf{0}_P, \gamma^{-1}\mathbf{I}_P)$:

$$
\begin{aligned}
\log p(\theta|S) &= \log p(S|\theta) + \log p(\theta) - \log Z \\
&= -L(S, \theta) + \log p(\theta) - \log Z \\
&\approx -L(S, \theta^*) - (\theta - \theta^*)^\top (H + \gamma \mathbf{I}_P)(\theta - \theta^*)/2 - \log Z \\
&\propto -(\theta - \theta^*)^\top (H + \gamma \mathbf{I}_P)(\theta - \theta^*)/2
\end{aligned}
$$

Here, the training loss represents the negative log-likelihood $L(S, \theta) = -\log p(S|\theta)$, and $Z := \int p(\theta)p(S|\theta)d\theta$ represents the evidence in Bayesian inference [5]. Similar to the IF, LA becomes computationally intractable when dealing with modern architectures due to the complexity of the Hessian matrix. To address this computational challenge, recent works have proposed various sub-curvature approximations, such as KFAC [50] and sub-network [11], which provide computationally efficient alternatives for working with LA.

According to (11), samples with a high degree of self-influence experience significant (linearized) loss changes for the parameters sampled from LA. Furthermore, Theorem 4.1 reveals that the gradients, the origin of distributional bias, arise from the linearizations in (11). Hence, to address the distributional bias of the bilinear IF approximations in Proposition 3.1, we remove the linearizations in (11). This leads us to consider a modified version of $\mathcal{I}$, named $\mathcal{I}_{\text{LA}}$:

$$
\mathcal{I}_{\text{LA}}(z, z') := \mathbb{E}_{\psi \sim p_{\text{LA}}} \left[ \Delta \ell_{\theta^*}(z, \psi) \cdot \Delta \ell_{\theta^*}(z', \psi) \right]. \tag{12}
$$

The pseudocode for computing $\mathcal{I}_{\text{LA}}$ using Kronecker-Factored Approximate Curvature (KFAC; [40]) is available in Appendix C.

While $\mathcal{I}_{\text{LA}}$ no longer theoretically produces unimodal self-influence, it turns out that it still mixes correct and mislabeled samples, even in toy datasets. To illustrate this problem, we train a fully-connected NN on the two-circle dataset [29] with a modification as shown in Fig. 2(a): We add ten influential training samples at the center of the two-circle dataset containing 30 train samples per class (circle). These influential samples are highly susceptible (i.e., hard to recover) to leave-one-out retraining as samples of the opposite class are densely located around them. We provide other experimental details in Appendix E.

As discussed in Sec. 3, highly influential samples form a separate mode of high self-influence in the histogram of $\mathcal{I}_{\text{LOO}}$ (Fig. 2(b)) and $\mathcal{I}$ mixes typical and influential samples (Fig. 2(c)) due to the distributional bias of bilinear form. While Proposition 3.1 does not apply to $\mathcal{I}_{\text{LA}}$, Fig. 3 shows that $\mathcal{I}_{\text{LA}}$ still fails to distinguish between correct and mislabeled samples by severely overestimating the self-influence of typical samples (Fig. 3(a)-3(b)) or squeeze them into a unimodal distribution (Fig. 3(c)), similar to $\mathcal{I}$ (Fig. 3(d)).

### 4.2 Pitfalls of inverse Hessian in non-linear IF approximation

To analyze the limitation of $\mathcal{I}_{\text{LA}}$, we reparameterize the sampled parameter from LA as follows

$$
\psi = \theta^* + H^{-1/2}v = \theta^* + \sum_{i=1}^{P} \frac{v_i}{\sqrt{\lambda_i}} u_i \tag{13}
$$

From this reparameterization, one can see that for (13) to be valid, all eigenvalues of $H$ must be positive definite (i.e., all eigenvalues are positive), as Koh and Liang [25] assumed. However, over-parameterized NNs in practice can contain many zero eigenvalues in their Hessian.

**Proposition 4.2** (Hessian singularity for over-parameterized NNs)**.** *Let us assume a pre-trained parameter $\theta^* \in \mathbb{R}^P$ achieves zero training loss $L(S, \theta^*) = 0$ for squared error. Then, $H$ has at least $P - NK$ zero-eigenvalues for NNs such that $NK < P$. Furthermore, if $x$ is an input of training sample $z \in S$, then the following holds for the eigenvectors $\{u_i\}_{i=NK+1}^P$*

$$g_z^\top u_i = \nabla_{\hat{y}}^\top \ell(z, \theta^*) \underbrace{\nabla_\theta^\top f(x, \theta^*) u_i}_{\mathbf{0}_K} = 0 \tag{14}$$

We refer to Appendix B for the cross-entropy version of Proposition 4.2 with empirical Fisher (EF) matrix, a well-known approximation of the Hessian [31, 22], defined as $F := \frac{1}{N} \sum_{n=1}^N g_z g_z^\top$. Note that the over-parameterization assumption ($NK < P$) in Proposition 4.2 is prevalent in real-world situations, including the settings of Fig. 1- 3. Also, the zero training loss assumption can be satisfied with sufficient over-parameterization [14, 2]. Accordingly, empirical evidences of Proposition 4.2 have been reported in different scenarios [51, 18].

A simple solution to mitigate this singularity is adding an isotropic matrix to $H$, known as the **damping technique**: $H \approx H(\alpha) := H + \alpha \mathbf{I}_P$ for $\alpha > 0$. Then, the modified LA sample is

$$\psi_\alpha := \theta^* + (H(\alpha))^{-1/2} v = \theta^* + \sum_{i=1}^P \frac{v_i}{\sqrt{\lambda_i + \alpha}} u_i. \tag{15}$$

Although the damping trick can make all eigenvalues positive in principle, the order of the eigenvalues remains unchanged. Therefore, the null space components ($g_{z,i}$ for $i = NK + 1, \ldots, P$) in Proposition 4.2 are the most heavily weighted by $1/\sqrt{\alpha}$. These heavily weighted null space components do not affect the linearized sample-loss deviations in (11) as follows

$$\Delta \ell_{\theta^*}^{\mathrm{lin}}(z, \psi_\alpha) = g_z^\top (\psi_\alpha - \theta^*) = \sum_{i=1}^P \frac{v_i}{\sqrt{\lambda_i + \alpha}} g_z^\top u_i = \sum_{i=1}^{NK} \frac{v_i}{\sqrt{\lambda_i + \alpha}} g_z^\top u_i$$

by Proposition 4.2. However, this is not applicable to the sample-loss deviations in (12), since the null space components cause $\Delta \ell_{\theta^*}(z, \psi_\alpha)$ to change rapidly (i.e., overestimating the influence of samples). This can be confirmed by observing that setting $\alpha = 0.01$ in $\mathcal{I}_{\mathtt{LA}}$ (Fig. 3(a)) leads to significantly overestimating self-influences. To prevent this issue, one can enlarge $\alpha$ (i.e., decrease $1/\alpha$). However, since the scale of $\alpha$ does not change the order of eigenvalues, the null space components still receive a higher weight than the others. Consequently, $\mathcal{I}_{\mathtt{LA}}$ with $\alpha = 100$ (Fig. 3(b)) does not correctly represent the multi-modality of self-influence distribution.

Another way to handle the Hessian singularity is to limit (13) to only the top-$R$ eigenvalues of Hessian, similar to $\mathcal{I}_{\mathtt{Arnoldi}}$. However, this approximation method may result in significant errors due to using only the smallest $R$ eigenvalues of the inverse Hessian (i.e., $1/\sqrt{\lambda_i}$). Consequently, even linearized sample-loss deviations (i.e., $\mathcal{I}$) with $R = 50$ (Fig. 3(d)) suffer from severe performance degradation, compared to full Hessian with damping (Fig. 2(c)).

Compared to damping and truncation, $\mathcal{I}_{\mathtt{GEX}}$ (Fig. 2(d)) accurately captures the bimodal self-influence distribution. This effectiveness stems from the differential impact of SGD steps on typical and influential samples [61]: Since typical samples are robust to SGD steps [35], loss deviation of these samples are small in $\mathcal{I}_{\mathtt{GEX}}$. In contrast, influential samples experience significant loss deviations as they are sensitive to SGD steps. As GE makes diverse predictions [16], the members of GE introduce varying levels of loss deviation for each influential sample. An ablation study exploring the impact of ensemble size is provided in Appendix F.

### 4.3 Practical advantages of GEX

In addition to the above benefits, $\mathcal{I}_{\mathtt{GEX}}$ offers several implementation advantages. First, it can be obtained using **only open-source final checkpoints**, unlike $\mathcal{I}_{\mathtt{TracIn}}$ and $\mathcal{I}_{\mathtt{TracInRP}}$. This advantage broadens the range of applicable models. Second, batch estimation is easy to implement in $\mathcal{I}_{\mathtt{GEX}}$. Projection-based methods like $\mathcal{I}_{\mathtt{TracInRP}}$ and $\mathcal{I}_{\mathtt{Arnoldi}}$ require JVP computation, which is only efficient

Table 1: Area Under Curve (AUC) and Average Precision (AP) for noisy label detection tasks on four datasets. Due to the high time complexity associated with sample-wise gradients, we do not repeatedly measure the self-influence of $\mathcal{I}_{\text{TracIn}}$.

| | Synthetic label noise | | | | Real-world label noise | | | |
| | CIFAR-10 | | CIFAR-100 | | CIFAR-10-N | | CIFAR-100-N | |
| Detection method | AUC | AP | AUC | AP | AUC | AP | AUC | AP |
|---|---|---|---|---|---|---|---|---|
| Deep-KNN | 92.51 ± 0.19 | 69.93 ± 0.71 | 84.00 ± 0.14 | 40.17 ± 0.23 | 78.32 ± 0.19 | 72.60 ± 0.33 | 71.59 ± 0.21 | 59.76 ± 0.25 |
| CL | 57.60 ± 0.30 | 16.27 ± 0.20 | 84.16 ± 0.10 | 35.76 ± 0.50 | 75.94 ± 0.02 | 66.50 ± 0.09 | 69.49 ± 0.15 | 58.69 ± 0.23 |
| F-score | 73.34 ± 0.07 | 16.27 ± 0.09 | 59.18 ± 0.21 | 11.04 ± 0.05 | 69.39 ± 0.06 | 52.89 ± 0.06 | 68.95 ± 0.11 | 52.29 ± 0.14 |
| EL2N | 98.29 ± 0.03 | 95.82 ± 0.06 | 96.42 ± 0.05 | 73.82 ± 0.42 | 93.57 ± 0.17 | 91.26 ± 0.13 | 84.65 ± 0.08 | 77.26 ± 0.06 |
| $\mathcal{I}_{\text{RandProj}}$ | 62.70 ± 0.19 | 17.90 ± 0.17 | 79.96 ± 0.32 | 26.25 ± 0.47 | 56.75 ± 0.38 | 45.61 ± 0.38 | 67.25 ± 0.09 | 54.14 ± 0.09 |
| $\mathcal{I}_{\text{TracIn}}$ | 89.89 | 43.21 | 75.53 | 22.25 | 76,48 | 64.69 | 68.91 | 55.86 |
| $\mathcal{I}_{\text{TracInRP}}$ | 89.56 ± 0.14 | 44.26 ± 0.37 | 74.99 ± 0.25 | 21.62 ± 0.26 | 77.24 ± 0.45 | 65.17 ± 0.68 | 69.04 ± 0.28 | 56.41 ± 0.31 |
| $\mathcal{I}_{\text{Arnoldi}}$ | 61.64 ± 0.13 | 17.05 ± 0.18 | 77.20 ± 0.35 | 22.61 ± 0.42 | 56.83 ± 0.40 | 45.63 ± 0.40 | 66.57 ± 0.12 | 53.26 ± 0.11 |
| $\mathcal{I}_{\text{GEX-lin}}$ | 64.11 ± 0.34 | 18.34 ± 0.36 | 76.06 ± 0.36 | 22.26 ± 0.47 | 56.88 ± 0.29 | 45.67 ± 0.33 | 65.68 ± 0.15 | 52.66 ± 0.13 |
| $\mathcal{I}_{\text{GEX}}$ | **99.74 ± 0.02** | **98.31 ± 0.06** | **99.33 ± 0.03** | **96.08 ± 0.12** | **96.20 ± 0.03** | **94.89 ± 0.04** | **89.76 ± 0.01** | **86.30 ± 0.01** |

for packages that provide forward-mode auto-differentiation, such as JAX [6]. In contrast, batch computation in $\mathcal{I}_{\text{GEX}}$ necessitates **only forward computations**, which are efficient in almost all auto-differentiation packages. Since IF is computed for each training sample in most downstream tasks [25, 48, 52], efficient batching is critical for practical applications. As a result, we believe that this distinction will be vital for researchers and practitioners in their work. We provide the complexity analysis of $\mathcal{I}_{\text{GEX}}$ and other IF approximations in Appendix D and discuss the limitations of our method and broader impacts in Appendix G.

## 5 Experiments

Here we describe experiments demonstrating the usefulness of GEX in downstream tasks. We conducted two experiments for each noisy and clean label setting: Detection (Sec. 5.1) and relabeling (Sec. 5.2) of mislabeled examples for noisy label setting and dataset pruning (Sec. 5.3) and separating data sources (Sec. 5.4) for clean label setting. We refer to Appendix E for experimental details.

### 5.1 Noisy label detection

In this section, we evaluate the performance of IF approximation methods for detecting noisy labels. We use self-influence as an index for the noisy label of IF approximation methods following Koh and Liang [25] and Pruthi et al. [48]. A high degree of self-influence indicates mislabeled examples since the removal of mislabeled examples will significantly change prediction or loss.

We train ResNet-18 [20] on CIFAR-10/100 [27] with 10% random label corruption for synthetic noise and CIFAR-N [64] for real-world noise. We use the "worse label" version of CIFAR-10-N since it corresponds to the highest noise level. We compare $\mathcal{I}_{\text{GEX}}$ to the following baselines: $\mathcal{I}_{\text{TracInRP}}$ with 5 checkpoints and 20 random projections, $\mathcal{I}_{\text{RandProj}}$ with 20 random projections only for the final checkpoint, $\mathcal{I}_{\text{Arnoldi}}$ with 100 iterations and 20 projections. We assess F-score [61], and EL2N [46] as they offer alternative approaches to identifying influential samples. We also evaluate an ablative version of $\mathcal{I}_{\text{GEX}}$: $\mathcal{I}_{\text{GEX-lin}}$, which replaces LA with GE for (11) (i.e., $\mathcal{I}_{\text{GEX}}$ with linear sample-loss deviation). Finally, we report the performance of well-known baselines, Deep-KNN [4] and CL [43], for comparison. We provide results for cross-influence $\mathcal{I}(z, z')$ in case of $z \neq z'$ in Appendix F.

Table 1 shows that $\mathcal{I}_{\text{GEX}}$ distinguishes mislabeled samples better than other methods. Furthermore, we find that $\mathcal{I}_{\text{GEX}}$ is more effective than CL, a state-of-the-art noisy label detection technique. Since CL requires additional $K$-fold cross-validation for sample scoring ($K = 2$ used in our experiments), $\mathcal{I}_{\text{GEX}}$ will be an attractive alternative in terms of both computational complexity and detection performance. Notably, results of $\mathcal{I}_{\text{GEX-lin}}$ demonstrate that removing linearization is essential for performance improvement of $\mathcal{I}_{\text{GEX}}$.

One interesting observation in Table 1 is that $\mathcal{I}_{\text{Arnoldi}}$ did not show improvement compared to simple $\mathcal{I}_{\text{RandProj}}$. Indeed, a similar observation was reported in Table 1 in Schioppa et al. [52]. In contrast, $\mathcal{I}_{\text{TracInRP}}$ showed improvements in CIFAR-10 and SVHN compared to $\mathcal{I}_{\text{RandProj}}$, $\mathcal{I}_{\text{TracInRP}}$ is computationally expensive since it requires JVP computations for multiple checkpoints. On the other hand, $\mathcal{I}_{\text{GEX}}$ can be applied in a pure *post-hoc* style and does not require (framework-dependent)

Table 2: Noisy label detection performance for ImageNet [12]

| Detection method | ViT-S-32 | | MLP-Mixer-S-32 | |
|---|---|---|---|---|
| | AUC | AP | AUC | AP |
| F-score | 1.56 | 9.44 | 1.07 | 9.21 |
| EL2N | 87.28 | 38.37 | 88.47 | 41.63 |
| $\mathcal{I}_{\texttt{RandProj}}$ | 76.09 | 12.08 | 82.36 | 21.86 |
| $\mathcal{I}_{\texttt{TracInRP}}$ | 72.22 | 15.09 | 75.01 | 16.62 |
| $\mathcal{I}_{\texttt{Arnoldi}}$ | 73.95 | 16.09 | 82.16 | 21.83 |
| $\mathcal{I}_{\texttt{GEX}}$ | **99.39** | **95.22** | **98.73** | **90.65** |

Table 3: Relabeled test accuracy for mislabeled samples

| Detection method | Synthetic label noise | | Real-world label noise | |
|---|---|---|---|---|
| | CIFAR-10 | CIFAR-100 | CIFAR-10-N | CIFAR-100-N |
| Clean label acc. | $95.75 \pm 0.06$ | $79.08 \pm 0.05$ | $95.75 \pm 0.06$ | $79.08 \pm 0.05$ |
| Noisy label acc. | $90.94 \pm 0.15$ | $72.35 \pm 0.17$ | $68.63 \pm 0.32$ | $55.50 \pm 0.09$ |
| Detection method | Relabeled acc. | | | |
| Deep-KNN | $91.58 \pm 0.10$ | $66.12 \pm 0.27$ | $69.12 \pm 0.25$ | $50.03 \pm 0.19$ |
| CL | $91.11 \pm 0.10$ | $72.55 \pm 0.13$ | $30.52 \pm 0.02$ | $33.17 \pm 0.02$ |
| F-score | $78.94 \pm 0.39$ | $58.67 \pm 0.18$ | $53.50 \pm 0.28$ | $44.34 \pm 0.21$ |
| EL2N | $89.40 \pm 0.10$ | $61.72 \pm 0.18$ | $72.01 \pm 0.51$ | $47.58 \pm 0.22$ |
| $\mathcal{I}_{\texttt{RandProj}}$ | $90.94 \pm 0.09$ | $72.42 \pm 0.16$ | $68.55 \pm 0.17$ | $55.47 \pm 0.08$ |
| $\mathcal{I}_{\texttt{TracIn}}$ | $91.24$ | $72.07$ | $68.36$ | $54.87$ |
| $\mathcal{I}_{\texttt{TracInRP}}$ | $90.82 \pm 0.06$ | $71.70 \pm 0.15$ | $68.12 \pm 0.23$ | $55.20 \pm 0.06$ |
| $\mathcal{I}_{\texttt{Arnoldi}}$ | $91.10 \pm 0.09$ | $72.50 \pm 0.08$ | $68.67 \pm 0.02$ | $55.37 \pm 0.14$ |
| $\mathcal{I}_{\texttt{GEX-lin}}$ | $91.04 \pm 0.16$ | $70.08 \pm 0.12$ | $68.44 \pm 0.08$ | $55.51 \pm 0.21$ |
| $\mathcal{I}_{\texttt{GEX}}$ | $\mathbf{93.54 \pm 0.05}$ | $\mathbf{75.04 \pm 0.10}$ | $\mathbf{73.94 \pm 0.24}$ | $\mathbf{57.13 \pm 0.10}$ |

JVP computation. Motivated by $\mathcal{I}_{\texttt{GEX}}$, one can propose a purely post-hoc $\mathcal{I}_{\texttt{TracInRP}}$ using checkpoints generated by GE. We provide an ablation study for this setting in Appendix F.

To verify the scalability of results, we train Vision Transformer (ViT; [13]) and MLP-Mixer [60] on ImageNet [12] with 10% label corruption and evaluate the performance of IF approximation methods. Table 2 shows that the performance gap between $\mathcal{I}_{\texttt{GEX}}$ and other baselines is still large. Specifically, in this scenario, F-score fails since some samples have never been correctly predicted during pre-training (i.e., no forgetting events for such samples). We provide additional results for other vision datasets (MNIST [33] and SVHN [42]) and text classification settings in Appendix F.

## 5.2  Relabeling mislabeled samples

Following up on the detection task in Sec. 5.1, we improve classification performance by relabeling mislabeled samples following Kong et al. [26]. To this end, Otsu algorithm [44] was used to find a threshold that distinguishes between noisy and clean labels, given the self-influence of approximation methods. Since the Otsu method has no inputs other than (influence) distribution, practitioners can apply it for noisy label settings without onerous hyperparameter optimization. Following Kong et al. [26], we relabel training samples as follows:

$$\tilde{y}_k = \begin{cases} 0, & \text{if } k = m, \\ \log_{\varphi_k} \sqrt[\kappa-1]{1 - \varphi_m}, & \text{otherwise} \end{cases} \qquad (16)$$

where $m$ is the training sample's original (corrupted) label and $\varphi_i$ is the predicted probability for $i$-th class of the training sample. Therefore, the relabel function in (16) masks the original (noisy) label and re-distributes the remaining probabilities. We provide additional results for MNIST and SVHN in Appendix F.

Table 3 shows the relabeled test accuracy with the other settings for comparison. A critical observation in Table 3 is that not all methods achieve performance improvements with relabeling. Otsu algorithm did not find an appropriate threshold for mislabeled examples for these methods. In contrast, since $\mathcal{I}_{\texttt{GEX}}$ can separate mislabeled samples from the self-influence distribution, performance improvements were obtained through relabeling. We provide additional results for ImageNet in Appendix F.

## 5.3  Dataset pruning

We evaluate methods in the previous section on data pruning task [58] to validate the efficacy of $\mathcal{I}_{\texttt{GEX}}$ in clean label settings. We quantify the importance of training samples based on self-influence following Feldman and Zhang [15]: We prune low self-influence samples as they are relatively easy samples that can be generalized by learning other training samples. We use networks and datasets in Sec. 5.1 without label noise. We provide additional results for MNIST and SVHN in Appendix F.

As shown in Fig. 4, $\mathcal{I}_{\texttt{GEX}}$ consistently detects samples that can be pruned. It is worth noting that among the IF approximation methods considered, only $\mathcal{I}_{\texttt{GEX}}$ is comparable to the well-known state-of-the-art methods, F-score and EL2N. This suggests that existing IF approximation methods may experience a performance decrease in downstream tasks because of the errors caused by distributional bias.

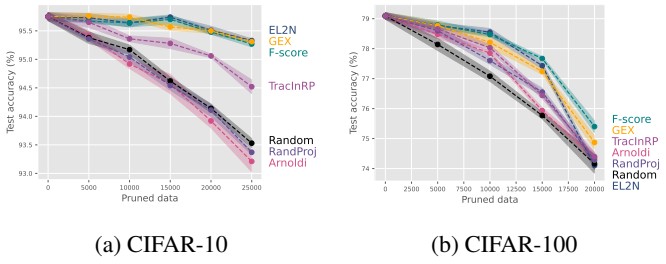

|            | (a) CIFAR-10 | (b) CIFAR-100 |

**Table 4:** Detection metrics for separating MNIST and SVHN

| Detection method | AUC | AP |
|---|---|---|
| Deep-KNN | 44.58 ± 0.39 | 48.13 ± 0.27 |
| CL | 49.88 ± 0.15 | 49.91 ± 0.11 |
| F-score | 67.39 ± 0.02 | 65.37 ± 0.02 |
| EL2N | 68.36 ± 0.33 | 72.20 ± 0.24 |
| $\mathcal{I}_{\texttt{RandProj}}$ | 46.88 ± 1.22 | 56.53 ± 0.82 |
| $\mathcal{I}_{\texttt{TracInRP}}$ | 56.10 ± 1.83 | 61.53 ± 1.65 |
| $\mathcal{I}_{\texttt{Arnoldi}}$ | 38.33 ± 0.23 | 50.69 ± 0.11 |
| $\mathcal{I}_{\texttt{GEX-lin}}$ | 53.69 ± 1.52 | 57.88 ± 1.06 |
| $\mathcal{I}_{\texttt{GEX}}$ | **73.11 ± 0.73** | **74.42 ± 0.46** |

**Figure 4:** Data pruning results using various scoring methods. We refer to Sec. 5.3 for the details of scoring methods.

## 5.4 Separation of data sources

We also evaluate IF approximations on separating heterogeneous data sources following Harutyunyan et al. [19]. One often combines multiple datasets to improve generalization. In general, these datasets differ in their informativeness. Therefore, IF can be used to determine which dataset is the most essential based on the informativeness of each dataset. In this experiment, we train ResNet-18 on a mixed dataset of MNIST and SVHN, with 25K random subsamples for each training dataset. Then, we use self-influence as an index for the informativeness following Harutyunyan et al. [19].

Table 4 shows detection metrics for MNIST and SVHN separation using various IF approximations (and other baselines in previous sections). In contrast to the noisy label detection experiment in Sec. 5.1, many IF approximations in this experiment fail to distinguish between data sources: MNIST and SVHN can only be significantly separated by $\mathcal{I}_{\texttt{GEX}}$, $\mathcal{I}_{\texttt{TracInRP}}$, and F-score. Assuming only the final checkpoint is available, only $\mathcal{I}_{\texttt{GEX}}$ can be applied among them.

## 6 Conclusion

In this work, we studied the oversimplification of influence distributions due to their bilinear approximations. To mitigate this bias, we developed a non-linear IF approximation, GEX, with GE. Empirically, GEX consistently outperforms the standard IF approximations in various downstream tasks in practice. Also, GEX is user-friendly as it only requires the final checkpoint of pretraining and excludes JVP computation, often constrained by frameworks, for efficient batch estimation. With these advantages, GEX can be used as a practical tool for researchers and practitioners.

## Acknowledgements

This work was supported by the Institute of Information & Communications Technology Planning & Evaluation (IITP) grant funded by the Korea government (MSIT) (No.2019-0-00075 / No.2022-0-00984) and the National Research Foundation of Korea (NRF) grants (No.2018R1A5A1059921 / RS-2023-00209060) funded by the Korea government (MSIT). This work was also supported by Samsung Electronics Co., Ltd (No.IO201214-08133-01).

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

# A   Notations

Table 5: Notations used in the main paper

| | | |
|---:|:---:|:---|
| $\mathcal{D}$ | $\triangleq$ | (true) data distribution |
| $S := \{z_n : (x_n, y_n)\}_{n=1}^N$ | $\triangleq$ | i.i.d. sampled training dataset from $\mathcal{D}$ |
| $z_n := (x_n, y_n)$ | $\triangleq$ | a training instance (or sample) |
| $x_n \in \mathbb{R}^D, y_n \in \mathbb{R}^K$ | $\triangleq$ | training inputs/outputs |
| $\theta, \psi \in \mathbb{R}^P$ | $\triangleq$ | parameter of NNs |
| $\hat{y}_n = f(x_n, \theta)$ | $\triangleq$ | output of NN given $x, \theta$ |
| | | |
| $\ell(z, \theta)$ | $\triangleq$ | instance-wise loss given sample $z, \theta$ |
| $L(S, \theta)$ | $\triangleq$ | $\frac{1}{N} \sum_{n=1}^N \ell(z_n, \theta)$ |
| $\theta^* \in \mathbb{R}^P$ | $\triangleq$ | pre-trained parameter of NN |
| $\theta_z^* \in \mathbb{R}^P$ | $\triangleq$ | $\mathrm{argmin}_\theta\, L(S, \theta) - \ell(z, \theta)/N$ (leave-one-out retrained parameter) |
| $g_z \in \mathbb{R}^P$ | $\triangleq$ | $\nabla_\theta \ell(z, \theta^*)$ (gradient of instance-loss w.r.t. parameter given $z, \theta^*$) |
| $H \in \mathbb{R}^{P \times P}$ | $\triangleq$ | $\nabla_\theta^2 L(\mathcal{S}, \theta^*)$ (Hessian of training loss w.r.t. parameter given $\theta^*$) |
| $\ell_{\theta^*}^{\mathrm{lin}}(z, \psi)$ | $\triangleq$ | $\ell(z, \theta^*) + g_z^\top (\psi - \theta^*)$ (linearization of instance-loss w.r.t. parameter) |
| $\Delta \ell_{\theta^*}^{\mathrm{lin}}(z, \psi)$ | $\triangleq$ | $\ell_{\theta^*}^{\mathrm{lin}}(z, \psi) - \ell(z, \theta^*)$ |
| $\Delta \ell_{\theta^*}(z, \psi)$ | $\triangleq$ | $\ell_{\theta^*}(z, \psi) - \ell(z, \theta^*)$ |
| $\mathbf{0}_P \in \mathbb{R}^P$ | $\triangleq$ | the $P$-dimensional zero vector |
| $\mathbf{I}_P \in \mathbb{R}^{P \times P}$ | $\triangleq$ | the $P$-dimensional identity matrix |
| | | |
| $g_z^c \in \mathbb{R}^P$ | $\triangleq$ | $\nabla_\theta \ell(z, \theta^c)$ (gradient of instance-loss w.r.t. parameter for a checkpoint $c$) |
| $Q_R \in \mathbb{R}^{P \times R}$ | $\triangleq$ | Random matrix sampled from $\mathcal{N}(0, 1/R)$ component-wisely |
| $H = U \Lambda U^\top$ | $\triangleq$ | Eigendecomposition of Hessian |
| $H(\alpha)$ | $\triangleq$ | Damped Hessian matrix $H + \alpha \mathbf{I}_P$ |
| $\Lambda_R \in \mathbb{R}^{R \times R}, U_R \in \mathbb{R}^{P \times R}$ | $\triangleq$ | Submatrices of the top-$D$ eigenvalues/eigenvectors |
| $\lambda_i \in \mathbb{R}, u_i \in \mathbb{R}^P$ | $\triangleq$ | the $i$-th largest eigenvalue of $H$ and its eigenvector |
| $g_{z,i} = g_z^\top u_i \in \mathbb{R}$ | $\triangleq$ | $i$-th component of $g_z$ in the eigenspace of $H$ |
| | | |
| $p_{\mathtt{LA}}(\psi)$ | $\triangleq$ | $\mathcal{N}(\psi | \theta^*, H^{-1})$ (the Laplace approximated posterior) |
| $p_{\mathtt{GE}}(\psi)$ | $\triangleq$ | $\frac{1}{M} \sum_{m=1}^M \delta_{\theta^{(m)}}(\psi)$ (the Geometric Ensemble distribution) |
| $\{\theta^{(m)}\}_{m=1}^M$ | $\triangleq$ | checkpoints of Geometric Ensemble (GE) |

# B  Proofs

## B.1  Proof of Proposition 3.1

**Proposition B.1** (Distributional bias in bilinear self-influence)**.** *Let us assume $g_z$ follows a $P$-dimensional stable distribution (e.g., Gaussian, Cauchy, and Lévy distribution) and $M \in \mathbb{R}^{P \times P}$ is a positive (semi-)definite matrix. Then, self-influence in the form of $\mathcal{I}_M(z, z) = g_z^\top M g_z$ follows a unimodal distribution. Furthermore, if $g_z$ follows a Gaussian distribution, then the self-influence follows a generalized $\chi^2$-distribution.*

*Proof.* We first consider the case of gradients with stable distributions [56]. According to Corollary 1 in Mittnik et al. [41], the sum of squares of stable random variables is a stable distribution with the stability of $\alpha/2$. Since stable distributions are unimodal [66, 54], the resulting self-influence is also unimodal.

Now we proof the case of the Gaussian gradient. By positive semi-definite assumption, let us write the eigendecomposition of $M$ as $M = \sum_{i=1}^{P} \lambda_i u_i u_i^\top$ where eigenvalues are sorted in descending order. Specifically, one can compress this summation for the strictly positive eigenvalues: $M = \sum_{i=1}^{T} \lambda_i u_i u_i^\top$ where $T$ is the cardinality of strictly positive eigenvalues. Then the self-influence $\mathcal{I}_M$ can be expressed as follows:

$$\mathcal{I}_M(z, z) = g_z^\top M g_z = \sum_{i=1}^{T} \lambda_i g_{z,i}^2$$

where $g_{z,i} = g_z^\top u_i$ is the $i$-th component of $g_z$ in the eigenspace of $M$. As a linearly transformed Gaussian random variable follows Gaussian distribution [5], one can write $g_{z,i}$ as follows

$$g_{z,i} \sim \mathcal{N}(g_{z,i} \mid \mu_i, \sigma_i^2).$$

Therefore, $g_{z,i}$ follows following generalized $\chi^2$-distribution [8]

$$g_{z,i}^2 \sim \tilde{\chi}^2(g_{z,i}^2 \mid \sigma_i^2, 1, \mu_i^2).$$

where $\mu_i^2$ is the non-centrality parameter and $\sigma_i^2$ is the weight of non-central $\chi^2$. As a result, $\mathcal{I}_M$ follows following generalized $\chi^2$-distribution

$$\mathcal{I}_M(z, z) \sim \tilde{\chi}^2(\mathcal{I}_M(z, z) \mid \sigma^2, \mathbf{1}_T, \mu^2).$$

where $\mu^2 = (\mu_1^2, \ldots, \mu_T^2) \in \mathbb{R}^T$, $\sigma^2 = (\sigma_1^2, \ldots, \sigma_T^2) \in \mathbb{R}^T$, and $\mathbf{1}_T = (1, \ldots, 1) \in \mathbb{R}^T$. $\qquad\square$

## B.2  Proof of Theorem 4.1

**Theorem B.2** (Connection between IF and LA)**.** *$\mathcal{I}$ in Koh and Liang [25] can be expressed as*

$$\mathcal{I}(z, z') = \mathbb{E}_{\psi \sim p_{\text{LA}}} \left[ \Delta \ell_{\theta^*}^{\text{lin}}(z, \psi) \cdot \Delta \ell_{\theta^*}^{\text{lin}}(z', \psi) \right] \tag{17}$$

*where $\Delta \ell_{\theta^*}^{\text{lin}}(z, \psi) := \ell_{\theta^*}^{\text{lin}}(z, \psi) - \ell_{\theta^*}^{\text{lin}}(z, \theta^*) = g_z^\top (\psi - \theta^*)$ and $p_{\text{LA}}$ is the Laplace approximated posterior*

$$p_{\text{LA}}(\psi) := \mathcal{N}\left(\psi | \theta^*, H^{-1}\right).$$

*Proof.* One can express $\Delta \ell_{\theta^*}^{\text{lin}}(z, \psi)$ as follows

$$\Delta \ell_{\theta^*}^{\text{lin}}(z, \psi) = \ell_{\theta^*}^{\text{lin}}(z, \psi) - \ell_{\theta^*}^{\text{lin}}(z, \theta^*) = g_z^\top (\psi - \theta^*)$$

Therefore, $\mathbb{E}_{\psi \sim p_{\text{LA}}} \left[ \Delta \ell_{\theta^*}^{\text{lin}}(z, \psi) \cdot \Delta \ell_{\theta^*}^{\text{lin}}(z', \psi) \right]$ can be arranged as follows

$$\begin{aligned} \mathbb{E}_{\psi \sim p_{\text{LA}}} \left[ \Delta \ell_{\theta^*}^{\text{lin}}(z, \psi) \cdot \Delta \ell_{\theta^*}^{\text{lin}}(z', \psi) \right] &= \mathbb{E}_{\psi \sim p_{\text{LA}}} \left[ g_z^\top (\psi - \theta^*)(\psi - \theta^*)^\top g_{z'} \right] \\ &= g_z^\top \mathbb{E}_{\psi \sim p_{\text{LA}}} \left[ (\psi - \theta^*)(\psi - \theta^*)^\top \right] g_{z'} \\ &= g_z^\top H^{-1} g_{z'} = \mathcal{I}(z, z'). \end{aligned}$$

$\square$

## B.3 Proof of Proposition 4.2

**Proposition B.3** (Singular Hessian for over-parameterized NNs). *Let us assume a pre-trained parameter $\theta^* \in \mathbb{R}^P$ achieves zero training loss $L(S, \theta^*) = 0$ for squared error. Then, $H$ has at least $P - NK$ zero-eigenvalues for NNs such that $NK < P$. Furthermore, if $x$ is an input of training sample $z \in S$, then the following holds for the eigenvectors $\{u_i\}_{i=NK+1}^P$*

$$g_z^\top u_i = \nabla_{\hat{y}}^\top \ell(z, \theta^*) \underbrace{\nabla_\theta^\top f(x, \theta^*) u_i}_{\mathbf{0}_K} = 0 \tag{18}$$

*Proof.* The Hessian matrix of training loss w.r.t. parameters can be expressed as follows [51]

$$
\begin{aligned}
H &= \frac{1}{N} \sum_{n=1}^N \nabla_\theta f(x_n, \theta^*) \nabla_{\hat{y}_n}^2 \ell(z_n, \theta^*) \nabla_\theta^\top f(x_n, \theta^*) + \nabla_{\hat{y}_n} \ell(z_n, \theta^*) \nabla_\theta^2 f(x_n, \theta^*) \\
&= \frac{1}{N} \sum_{n=1}^N \nabla_\theta f(x_n, \theta^*) \nabla_{\hat{y}_n}^2 \ell(z_n, \theta^*) \nabla_\theta^\top f(x_n, \theta^*) \\
&= \frac{1}{N} \sum_{n=1}^N \nabla_\theta f(x_n, \theta^*) \nabla_\theta^\top f(x_n, \theta^*)
\end{aligned}
$$

where the first equality holds due to the zero training loss assumption

$$\nabla_{\hat{y}_n} \ell(z_n, \theta^*) = (f(x_n, \theta^*) - y_n) = \mathbf{0}_K \quad \forall n = 1, \dots, N$$

and the second equality holds due to the squared error assumption

$$\nabla_{\hat{y}_n}^2 \ell(z_n, \theta^*) = \mathbf{I}_K \quad \forall n = 1, \dots, N.$$

Therefore, $H$ is a sum of $NK$ rank-one products. As a result, $H$ has at most $NK$ non-zero eigenvalues and at least $P - NK$ zero-eigenvalues for over-parameterized NNs such that $NK < P$. As training Jacobians $\nabla_\theta^\top f(x, \theta^*)$ for $n = 1, \dots, N$ are contained in the linear span of $\{u_i\}_{i=1}^{NK}$, which is orthogonal to $\{u_i\}_{i=NK+1}^P$, the following holds for any training input $x$

$$\nabla_\theta^\top f(x, \theta^*) u_i = \mathbf{0}_K, \quad \forall u_i \in \{u_i\}_{i=NK+1}^P.$$

$\square$

## B.4 Cross-entropy version of Proposition 4.2 and its proof

**Proposition B.4** (Cross-entropy version of Proposition 4.2). *Let us assume a pre-trained parameter $\theta^* \in \mathbb{R}^P$ for cross-entropy loss. Then, the empirical Fisher (EF) matrix $F = \frac{1}{N} \sum_{z \in S} g_z g_z^\top$ has at least $P - N$ zero-eigenvalues for NNs such that $N < P$. Furthermore, the following holds for the last $P - N$ eigenvectors $\{u_i\}_{i=N+1}^P$ and any training sample $z \in S$*

$$g_z^\top u_i = 0 \tag{19}$$

*Proof.* As $F$ is a sum of $N$ rank-one products. As a result, $F$ has at most $N$ non-zero eigenvalues and at least $P - N$ zero-eigenvalues for over-parameterized NNs such that $N < P$. As gradients $\nabla_\theta^\top \ell(z_n, \theta^*)$ for $n = 1, \dots, N$ are contained in the linear span of $\{u_i\}_{i=1}^N$, which is orthogonal to $\{u_i\}_{i=N+1}^P$, the following holds for any training input $x$

$$g_z u_i = 0, \quad \forall u_i \in \{u_i\}_{i=N+1}^P.$$

$\square$

# C  Implementations

## C.1  Pseudocode of LA for IF approximation

---

**Algorithm 1** $\mathcal{I}_{\text{LA}}$ with KFAC [39, 17] sub-curvature approximation

---

1: **Input**: training data $S$, pre-trained parameter $\theta^*$, number of LA samples $M$, two data samples $z, z'$
2:
3: # Estimation of KFAC sub-curvature
4: Initialize activation statistics $A^l$ and gradient statistics $G^l$ for each layer $l = 1, \ldots, L$
5: **for** $n = 1, \ldots, N$ (This computation can be conducted batch-wisely.) **do**
6:     **for** $l =, 1, \ldots, L$ **do**
7:         Accumulate activation statistics $A^l$ and gradient statistics $G^l$
8:     **end for**{End accumulation for layers}
9: **end for**{End accumulation for training samples}
10: Compute eigendecomposition of activation/gradient statistics:

$$A^l = U_{A^l} \Lambda_{A^l} U_{A^l}, \; G^l = U_{G^l} \Lambda_{G^l} U_{G^l}.$$

11:
12: # Sample Laplace approximated posteriors
13: **for** $m = 1, \ldots, M$ **do**
14:     Sample $P$-dimensional standard Gaussian random variable $v_m \sim \mathcal{N}(v \mid 0_P, I_P)$
15:     **for** $l =, 1, \ldots, L$ **do**
16:         Use reparameterization trick for each layer:

$$\psi_m^l \leftarrow (\theta^*)^l + U_{G^l} \left( v_m^l \odot \sqrt{(\text{diag}(\Lambda_{G^l}) \text{diag}(\Lambda_{A^l})^\top)} \right) U_{A^l}^\top$$

        where $\text{diag}(A)$ is the diagonal components of matrix $A$, $\odot$ denotes Hadamard product of two matrices, and the square root is applied for each component of the matrix.
17:     **end for**
18: **end for**{End LA sampling $\{\psi_m\}_{m=1}^M$}
19:
20: # Compute MC estimator of non-linear IF approximation (12)
21: $\hat{\mathcal{I}}_{\text{LA}}(z, z') \leftarrow \sum_{m=1}^M [\Delta \ell_{\theta^*}(z, \psi_m) \cdot \Delta \ell_{\theta^*}(z', \psi_m)]$
22:
23: **Output**: $\hat{\mathcal{I}}_{\text{LA}}(z, z')$

---

## C.2  Pseudocode of GEX

---

**Algorithm 2** $\mathcal{I}_{\text{GEX}}$

---

1: **Input**: training data $S$, pre-trained parameter $\theta^*$, number of LA samples $M$, number of fine-tuning steps $T$, two data samples $z, z'$
2:
3: # Generating Geometric Ensemble (GE; [16])
4: **for** $m = 1, \ldots, M$ (This computation can be parallelized for multiple devices) **do**
5:     Initialized the $m$-th checkpoint $\theta_m^0 \leftarrow \theta^*$ (or $\theta_{m-1}^T$)
6:     **for** $t = 1, \ldots, T$ **do**
7:         Apply stochastic optimization update (e.g., SGD with momentum): $\theta_m^t \leftarrow \theta_m^{t-1}$
8:     **end for**{End fine-tuning the $m$-th checkpoint $\theta_m^T$}
9: **end for**{End generation of GE $\{\theta_m^T\}_{m=1}^M$}
10:
11: # Compute the non-linear IF approximation (10)
12: $\hat{\mathcal{I}}_{\text{GEX}}(z, z') \leftarrow \sum_{m=1}^M [\Delta \ell_{\theta^*}(z, \theta_m^T) \cdot \Delta \ell_{\theta^*}(z', \theta_m^T)]$
13:
14: **Output**: $\hat{\mathcal{I}}_{\text{GEX}}(z, z')$

---

# D   Complexity analysis

Table 6: Comparison of computational complexity between IF approximations. $P$ and $P_{\mathtt{Last}}$ are the entire parameter dimension and the last-layer parameter dimension, respectively. $N$ is the number of training samples and $R$ is the projection dimension used in $\mathcal{I}_{\mathtt{Arnoldi}}$ and $\mathcal{I}_{\mathtt{TracInRP}}$. $C$ is the number of checkpoints used in $\mathcal{I}_{\mathtt{TracIn}}$ and $\mathcal{I}_{\mathtt{TracInRP}}$ and $T$ is the number of SGD steps used in $\mathcal{I}_{\mathtt{GEX}}$ (See Appendix C). $M_{\mathtt{GE}}$ and $M_{\mathtt{LA}}$ are the number of samples used in Geometric Ensemble and Laplace approximation, respectively. $C_{\mathtt{forward}}$, $C_{\mathtt{backward}}$, and $C_{\mathtt{jvp}}$ are the constant for the single computation of forward, backward, and JVP. $C_{\mathtt{forward}}$ and $C_{\mathtt{jvp}}$ are similar for packages that offer forward-mode AD computation and $C_{\mathtt{backward}}$ is much slower than the others.

| | Time complexity for cache | Space complexity for cache | Time complexity for evaluation |
|---|---|---|---|
| $\mathcal{I}_{\mathtt{LiSSA}}$ | $O(N_{\mathtt{LiSSA}} \cdot C_{\mathtt{backward}} \cdot P \cdot N)$ | $O(P \cdot N)$ | $O(C_{\mathtt{backward}} \cdot P)$ |
| $\mathcal{I}_{\mathtt{Last\text{-}Layer}}$ | $O(N \cdot P_{\mathtt{Last}}^3)$ | $O(P_{\mathtt{Last}}^2)$ | $O(C_{\mathtt{backward}} \cdot P)$ |
| $\mathcal{I}_{\mathtt{TracIn}}$ | Requires intermediate ckpts during pre-training | $O(C \cdot P)$ | $O(C \cdot C_{\mathtt{backward}} \cdot P)$ |
| $\mathcal{I}_{\mathtt{TracInRP}}$ | Requires intermediate ckpts during pre-training | $O(C \cdot P)$ | $O(C \cdot R \cdot C_{\mathtt{jvp}} \cdot P)$ |
| $\mathcal{I}_{\mathtt{Arnoldi}}$ | $O(N_{\mathtt{Arnoldi}} \cdot C_{\mathtt{backward}} \cdot P)$ | $O(N_{\mathtt{Arnoldi}} \cdot P)$ | $O(R \cdot C_{\mathtt{jvp}} \cdot P)$ |
| $\mathcal{I}_{\mathtt{LA}}$ with KFAC | $O(C_{\mathtt{KFAC}} \cdot P \cdot N)$ | $O(M_{\mathtt{LA}} \cdot P)$ | $O(M_{\mathtt{LA}} \cdot C_{\mathtt{forward}} \cdot P)$ |
| $\mathcal{I}_{\mathtt{GEX\text{-}lin}}$ | $O(T \cdot M_{\mathtt{GE}} \cdot C_{\mathtt{backward}} \cdot P)$ | $O(M_{\mathtt{GE}} \cdot P)$ | $O(M_{\mathtt{GE}} \cdot R \cdot C_{\mathtt{jvp}} \cdot P)$ |
| $\mathcal{I}_{\mathtt{GEX}}$ | $O(T \cdot M_{\mathtt{GE}} \cdot C_{\mathtt{backward}} \cdot P)$ | $O(M_{\mathtt{GE}} \cdot P)$ | $O(M_{\mathtt{GE}} \cdot C_{\mathtt{forward}} \cdot P)$ |

In Table 6, we compare computational complexities of various IF approximations, including $\mathcal{I}_{\mathtt{GEX}}$. We provide three computational complexities of each IF approximation method: time and space complexity for cache construction and the time complexity for evaluation. As discussed in Sec. 4.3, all IF approximations except $\mathcal{I}_{\mathtt{TracIn}}$ and $\mathcal{I}_{\mathtt{TracInRP}}$ can be implemented using only the final checkpoint, which is usually publicly released. However, the time and space complexity of cache construction are significantly different between the post-hoc methods: As mentioned in Sec. 2, the time and space complexity of $\mathcal{I}_{\mathtt{LiSSA}}$ for cache construction are proportional to the size of the training dataset $N$, as $\mathcal{I}_{\mathtt{LiSSA}}$ requires IHVP computation for each training sample. Therefore, $\mathcal{I}_{\mathtt{LiSSA}}$ cannot be used in datasets with many training samples. On the other hand, $\mathcal{I}_{\mathtt{Last\text{-}Layer}}$ requires high space complexity proportional to the square of the number of last-layer parameters $P_{\mathtt{Last}}$. As a result, $\mathcal{I}_{\mathtt{Last\text{-}Layer}}$ cannot be applied to classification tasks with many classes (e.g., ImageNet [12]). While $\mathcal{I}_{\mathtt{Arnoldi}}$ avoid this issue by using the principal subspace of $H$, we found that they suffer from the OOM issue on GPU RAM, as eigendecomposition computation (the DISTILL procedure in Schioppa et al. [52]) requires the space complexity proportional to $N_{\mathtt{Arnoldi}}$, the number of steps in $\mathcal{I}_{\mathtt{Arnoldi}}$, which is about 100-200 in Schioppa et al. [52]. However, $\mathcal{I}_{\mathtt{GEX}}$ and $\mathcal{I}_{\mathtt{GEX\text{-}lin}}$ does not suffer from the high time and space complexity for cache construction, as the cache construction procedure of them is just post-hoc fine-tuning in Garipov et al. [16].

For the time complexity for evaluation, we assume each method computes a single influence pair (e.g., $\mathcal{I}(z, z')$ or $\mathcal{I}(z, z)$). According to Table 6, all methods except $\mathcal{I}_{\mathtt{LA}}$ and $\mathcal{I}_{\mathtt{GEX}}$ require either backward computation ($C_{\mathtt{backward}}$) or JVP computation ($C_{\mathtt{jvp}}$). Among them, methods that require backward computation ($\mathcal{I}_{\mathtt{LiSSA}}$, $\mathcal{I}_{\mathtt{Last\text{-}Layer}}$, and $\mathcal{I}_{\mathtt{TracIn}}$) do not support batch computation for multiple samples, as mentioned in Sec. 2. Therefore, computing self-influence for all training samples in practice is inefficient. On the other hand, methods that require JVP computation ($\mathcal{I}_{\mathtt{TracInRP}}$, $\mathcal{I}_{\mathtt{Arnoldi}}$, and $\mathcal{I}_{\mathtt{GEX\text{-}lin}}$) are efficient for packages that offer forward-mode AD computation (e.g., JAX [6]). On the contrary, $\mathcal{I}_{\mathtt{LA}}$ and $\mathcal{I}_{\mathtt{GEX}}$ do not suffer from these issues, as they require only forward computation.

# E   Experimental settings

In the main paper, we conducted empirical studies in four different settings. We provide details on these settings in this section. We use four random seeds to compute the standard errors for all experiments in the main paper. We use 8 NVIDIA RTX 3090 GPUs for all experiments.

**Two-circle classification task**   We use a modified two-circle classification task to study the side effects of damping and truncation in Sec. 4. We use `sklearn.datasets.make_circles` function to sample the training samples. We use 30 train samples for each class (circle) and add ten influential samples at the center. We use a two-layer fully connected NN with a hidden dimension of 200. We use `jax.hessian` function to compute the Hessian matrix of training loss and `np.linalg.eigh` to compute the principal subspace of $H^{-1}$.

**Detecting and relabeling of mislabeled samples** Noisy label settings [67] are used to study the distributional bias of the standard bilinear IF approximations. We use standard datasets in the computer vision domain: MNIST [33], CIFAR-10/100 [27], and SVHN [42]. For MNIST, we use VGGNet-B [55] with batch normalization [23] after each convolution layer, similar to pytorch implementation in https://pytorch.org/vision/main/models/vgg.html. We use ResNet-18 [20] for CIFAR and SVHN. We use random label corruption for 10% of randomly selected training samples for synthetic noise and CIFAR-N [64] for real-world label noise. We use a batch size of 1024 for all datasets. For all datasets, we use the cosine learning rate annealing [36] with a linear warm-up where the peak learning rate is 0.4, and the warm-up ratio (percentage of warm-up steps to training steps) is 10%. We use 0.0005 for the L2 regularization coefficient. We use 200 training epochs for all datasets.

To verify the scalability of our method, we also use ImageNet [12] with Vision Transformer [13] and MLP-Mixer [60]. We use the "S-32" setting due to the heavy computation of ImageNet. We use random label corruption for 10% of randomly selected training samples. We use a batch size 4096 and RandomResizedCrop, following Zhuang et al. [68]. We use the cosine learning rate annealing with a linear warm-up where the peak learning is 0.003 and the warm-up step is 10,000. We use AdamW [37] with 0.3 for the weight decay coefficient. We train 300 epochs for ImageNet, following Zhuang et al. [68].

To verify the effectiveness of our method in another modality, we train SWEM [53] to DBpedia [34], following Pruthi et al. [48]. However, Pruthi et al. [48] used this setting for a clean label case study, while we used it for noisy label detection with 10% label corruption. For text representation, we use SentencePiece [28] tokenizer and apply average pooling of word embeddings. We train 60 epochs using the AdamW [37] optimizer with a peak learning rate of 0.001 and a weight decay of 0.001. We performed a grid search on the search space mentioned in [53] to find these hyperparameters, aiming to achieve the 95.5% train accuracy mentioned in [48]. We provide our results in Appendix F. Deep-KNN was omitted because it took more than 15 hours to calculate for each random seed.

We use 20 dimensions for all projection-based methods (i.e., $\mathcal{I}_{\texttt{RandProj}}$, $\mathcal{I}_{\texttt{TracInRP}}$, $\mathcal{I}_{\texttt{Arnoldi}}$). We use 100 iterations for $\mathcal{I}_{\texttt{Arnoldi}}$. Five checkpoints, evenly sampled from the latter half of training epochs, are used for $\mathcal{I}_{\texttt{TracInRP}}$ and Deep-KNN. We use 10% of training samples to compute the neighborhoods in Deep-KNN. For CL, we use two-fold cross-validation. Thus, CL requires at least a separate pre-training level of computational cost. We use 20 initial training epochs for EL2N with five repetitions. We use 32 samples for $\mathcal{I}_{\texttt{LA}}$ and $\mathcal{I}_{\texttt{GEX}}$. We use a cosine learning rate of 0.05 with 800 steps for the fine-tuning procedure to generate GE. We provide an ablation study exploring the impact of ensemble size in Appendix F.

**Dataset pruning** We conducted a dataset pruning task in Paul et al. [46] to validate $\mathcal{I}_{\texttt{GEX}}$ for clean label settings. We use the same pre-training and post-hoc hyperparameters as noisy label settings for this task (except for the presence of mislabeled samples). Since the pruning capacity (i.e., the maximum number of samples to be pruned) of all datasets is not the same, we prune 40K samples for MNIST (within 1% test acc. degradation for all methods), 25K samples for CIFAR-10 and SVHN (within 0.5% test acc. degradation for F-score, EL2N, and $\mathcal{I}_{\texttt{GEX}}$), and 20K samples for CIFAR-100 (within 4% test ACC. degradation for $\mathcal{I}_{\texttt{GEX}}$, F-score, $\mathcal{I}_{\texttt{LA}}$).

**Separation of data sources** We also conducted a data source separation task in Harutyunyan et al. [19] to validate $\mathcal{I}_{\texttt{GEX}}$ for clean label settings. In this task, we train a ResNet-18 on a mixed dataset of MNIST and SVHN, with 25,000 random subsamples for each training dataset. The other pre-train and post-hoc hyperparameters are the same as for noisy labels.

# F  Additional results

In this section, we conducted additional experiments for completeness. To this end, we provide an ablation study for the ensemble size of $\mathcal{I}_{\texttt{GEX}}$ in Appendix F.1. We also provide additional results of Sec. 5 in Appendix F.2- F.4. In Appendix F.5, we report the noisy label detection performance with cross-influence. Finally, we study a purely post-hoc $\mathcal{I}_{\texttt{TracInRP}}$ by using GE in Appendix F.6.

## F.1 Ablation study for the ensemble size

Table 7: AUC and AP for noisy label detection with 10% label corruption

| | CIFAR-10 | | CIFAR-100 | |
|---|---|---|---|---|
| $M_{\text{GE}}$ | AUC | AP | AUC | AP |
| 32 | 99.74 ± 0.02 | 98.31 ± 0.06 | 99.33 ± 0.03 | 96.08 ± 0.12 |
| 28 | 99.73 ± 0.01 | 98.16 ± 0.06 | 99.32 ± 0.03 | 95.92 ± 0.12 |
| 24 | 99.72 ± 0.01 | 98.08 ± 0.07 | 99.29 ± 0.03 | 95.79 ± 0.11 |
| 20 | 99.70 ± 0.01 | 97.99 ± 0.06 | 99.25 ± 0.04 | 95.59 ± 0.12 |
| 16 | 99.68 ± 0.02 | 97.86 ± 0.07 | 99.2 ± 0.04 | 95.32 ± 0.11 |
| 12 | 99.64 ± 0.01 | 97.64 ± 0.07 | 99.12 ± 0.05 | 94.94 ± 0.12 |
| 8 | 99.57 ± 0.02 | 97.23 ± 0.11 | 98.96 ± 0.05 | 94.16 ± 0.11 |
| 4 | 99.40 ± 0.04 | 96.27 ± 0.18 | 98.75 ± 0.07 | 92.99 ± 0.18 |
| $\mathcal{I}_{\text{Arnoldi}}$ ($N_{\text{Arnoldi}} = 100, R = 20$) | 61.64 ± 0.13 | 17.05 ± 0.18 | 77.20 ± 0.35 | 22.61 ± 0.42 |
| $\mathcal{I}_{\text{TracInRP}}$ ($C = 5, R = 20$) | 89.56 ± 0.14 | 44.26 ± 0.37 | 74.99 ± 0.25 | 21.62 ± 0.26 |

To evaluate the effectiveness of the size of an ensemble in $\mathcal{I}_{\text{GEX}}$, we conduct an ablation study for the number of fine-tuned models $M_{\text{GE}}$ for the noisy label detection task in Sec. 5.1 with 10% label corruption. Table 7 shows that even a small number of checkpoints (about 8) can achieve better performance of GEX than other state-of-the-art IF approximations. While we used 32 checkpoints to ensure a low variance of results, small checkpoints - as small as 4 - may be sufficient in situations requiring a low computational cost.

## F.2 Results on MNIST and SVHN

Table 8: Noisy label detection performance on MNIST and SVHN

| | MNIST | | SVHN | |
|---|---|---|---|---|
| Detection method | AUC | AP | AUC | AP |
| Deep-KNN | 97.28 ± 0.13 | 90.34 ± 0.41 | 93.07 ± 0.06 | 77.33 ± 0.36 |
| CL | 87.31 ± 0.07 | 58.16 ± 0.26 | 59.18 ± 0.21 | 11.04 ± 0.05 |
| F-score | 96.67 ± 0.03 | 57.24 ± 0.13 | 85.38 ± 0.07 | 25.42 ± 0.07 |
| EL2N | **99.99 ± 0.00** | **99.90 ± 0.02** | 99.70 ± 0.01 | 96.33 ± 0.04 |
| $\mathcal{I}_{\text{RandProj}}$ | 70.90 ± 0.48 | 20.61 ± 0.46 | 58.82 ± 0.28 | 17.94 ± 0.19 |
| $\mathcal{I}_{\text{TracInRP}}$ | 97.22 ± 0.05 | 71.22 ± 0.54 | 91.52 ± 0.16 | 47.25 ± 0.77 |
| $\mathcal{I}_{\text{Arnoldi}}$ | 70.02 ± 0.44 | 19.88 ± 0.40 | 58.23 ± 0.48 | 17.71 ± 0.24 |
| $\mathcal{I}_{\text{GEX-lin}}$ | 91.67 ± 0.23 | 62.93 ± 0.75 | 61.59 ± 0.18 | 18.95 ± 0.17 |
| $\mathcal{I}_{\text{GEX}}$ | **99.99 ± 0.00** | **99.90 ± 0.02** | **99.74 ± 0.00** | **96.59 ± 0.06** |

Table 9: Relabeled test accuracy for mislabeled samples

| | MNIST | SVHN |
|---|---|---|
| Clean label acc. | 98.93 ± 0.02 | 97.50 ± 0.01 |
| Noisy label acc. | 95.21 ± 0.04 | 94.84 ± 0.10 |
| Detection method | Relabeled acc. | |
| Deep-KNN | 98.75 ± 0.04 | 95.15 ± 0.06 |
| CL | 64.57 ± 0.30 | 79.08 ± 0.05 |
| F-score | 97.75 ± 0.08 | 91.71 ± 0.10 |
| EL2N | **99.13 ± 0.04** | 95.86 ± 0.04 |
| $\mathcal{I}_{\text{RandProj}}$ | 97.78 ± 0.07 | 94.89 ± 0.06 |
| $\mathcal{I}_{\text{TracInRP}}$ | 98.08 ± 0.03 | 94.54 ± 0.06 |
| $\mathcal{I}_{\text{Arnoldi}}$ | 97.69 ± 0.07 | 94.84 ± 0.03 |
| $\mathcal{I}_{\text{GEX-lin}}$ | 95.37 ± 0.14 | 94.78 ± 0.11 |
| $\mathcal{I}_{\text{GEX}}$ | 98.89 ± 0.03 | **96.32 ± 0.03** |

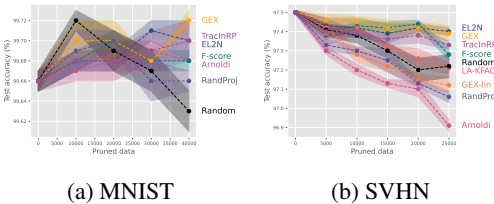

(a) MNIST   (b) SVHN

Figure 5: Data pruning results on MNIST and SVHN.

In this section, we provide the additional results of Sec. 5 for MNIST [33] and SVHN [42] with 10% label corruption. The overall tendency of MNIST and SVHN is similar to the results in Sec. 5. While EL2N is comparable (or even better) to $\mathcal{I}_{\text{GEX}}$ on MNIST, their good performance is not generalized to more scalable and practical settings in CIFAR-N [64], and ImageNet [12].

## F.3 Results on DBpedia

Table 10: Noisy label detection performance on DBpedia

| Detection method | DBPedia | |
| --- | --- | --- |
| | AUC | AP |
| CL | 86.49 ± 0.01 | 56.14 ± 0.04 |
| F-score | 45.18 ± 0.15 | 8.91 ± 0.01 |
| EL2N | 99.56 ± 0.00 | 95.26 ± 0.02 |
| $\mathcal{I}_{\texttt{RandProj}}$ | 98.42 ± 0.05 | 79.05 ± 0.90 |
| $\mathcal{I}_{\texttt{TracInRP}}$ | 98.42 ± 0.04 | 79.34 ± 0.60 |
| $\mathcal{I}_{\texttt{Arnoldi}}$ | 98.54 ± 0.02 | 80.69 ± 0.25 |
| $\mathcal{I}_{\texttt{GEX-lin}}$ | 97.10 ± 0.13 | 74.46 ± 1.14 |
| $\mathcal{I}_{\texttt{GEX}}$ | **99.79 ± 0.00** | **97.83 ± 0.03** |

In this section, we provide the additional results of Sec. 5.1 for DBPedia [34] with SWEM [53]. Table 10 shows that GEX better discriminates noisy labels than other methods. In addition to the robustness of various model types (including VGGNet, ResNet, ViT, and MLP-Mixer) and noise types (synthetic label noise and real-world label noise), this result indicates that GEX has consistently excellent performance for various data types.

## F.4 Results on 30% label corruption

Table 11: Noisy label detection performance with 30% label corruption

| Detection method | CIFAR-10 | | CIFAR-100 | |
| --- | --- | --- | --- | --- |
| | AUC | AP | AUC | AP |
| Deep-KNN | 88.39 ± 0.57 | 77.94 ± 1.04 | 82.09 ± 0.23 | 64.01 ± 0.37 |
| CL | 82.28 ± 0.04 | 64.55 ± 0.06 | 78.36 ± 0.13 | 60.42 ± 0.18 |
| F-score | 64.40 ± 0.21 | 31.98 ± 0.10 | 53.61 ± 0.12 | 28.91 ± 0.05 |
| EL2N | 99.13 ± 0.01 | 97.81 ± 0.05 | 97.50 ± 0.03 | 93.27 ± 0.10 |
| $\mathcal{I}_{\texttt{RandProj}}$ | 57.93 ± 0.51 | 33.36 ± 0.29 | 71.72 ± 0.01 | 46.33 ± 0.20 |
| $\mathcal{I}_{\texttt{TracInRP}}$ | 84.08 ± 0.56 | 59.56 ± 0.85 | 68.30 ± 0.29 | 43.79 ± 0.37 |
| $\mathcal{I}_{\texttt{Arnoldi}}$ | 57.26 ± 0.51 | 32.80 ± 0.28 | 69.34 ± 0.05 | 43.38 ± 0.23 |
| $\mathcal{I}_{\texttt{GEX-lin}}$ | 58.82 ± 0.46 | 33.93 ± 0.26 | 69.36 ± 0.25 | 43.96 ± 0.21 |
| $\mathcal{I}_{\texttt{GEX}}$ | **99.74 ± 0.01** | **98.19 ± 0.06** | **98.98 ± 0.01** | **97.84 ± 0.03** |

Table 12: Relabeled test accuracy for mislabeled samples

| | CIFAR-10 | CIFAR-100 |
| --- | --- | --- |
| Clean label acc. | 95.75 ± 0.06 | 79.08 ± 0.05 |
| Noisy label acc. | 79.81 ± 0.28 | 58.92 ± 0.04 |
| Detection method | Relabeled acc. | |
| Deep-KNN | 82.57 ± 0.43 | 56.26 ± 0.08 |
| CL | 35.26 ± 0.02 | 30.25 ± 0.65 |
| F-score | 65.29 ± 0.57 | 44.13 ± 0.40 |
| EL2N | 88.07 ± 0.17 | 60.86 ± 0.13 |
| $\mathcal{I}_{\texttt{RandProj}}$ | 79.80 ± 0.26 | 58.88 ± 0.19 |
| $\mathcal{I}_{\texttt{TracInRP}}$ | 79.59 ± 0.23 | 58.50 ± 0.28 |
| $\mathcal{I}_{\texttt{Arnoldi}}$ | 80.08 ± 0.13 | 58.76 ± 0.32 |
| $\mathcal{I}_{\texttt{GEX-lin}}$ | 79.72 ± 0.17 | 58.79 ± 0.10 |
| $\mathcal{I}_{\texttt{GEX}}$ | **89.88 ± 0.06** | **67.25 ± 0.07** |

In this section, we provide the additional results of Sec. 5.1-5.2 with 30% label corruption. Table 11-12 shows that $\mathcal{I}_{\texttt{GEX}}$ achieves better detection and relabeling performance than other methods on higher label noise settings. This is consistent with the empirical evaluation in Sec. 5 and shows the practicality of $\mathcal{I}_{\texttt{GEX}}$.

## F.5 Cross-influence results for noisy label detection

Table 13: AUC and AP for noisy label detection tasks on four datasets

| Detection method | MNIST | | CIFAR-10 | | CIFAR-100 | | SVHN | |
| --- | --- | --- | --- | --- | --- | --- | --- | --- |
| | AUC | AP | AUC | AP | AUC | AP | AUC | AP |
| $\mathcal{I}_{\texttt{RandProj}}$ | 56.64 ± 1.92 | 87.42 ± 0.61 | 58.72 ± 0.40 | 36.35 ± 0.56 | 51.59 ± 0.47 | 37.51 ± 0.39 | 59.97 ± 1.62 | 37.21 ± 1.45 |
| $\mathcal{I}_{\texttt{TracInRP}}$ | 57.09 ± 1.11 | 87.84 ± 0.31 | 60.74 ± 0.77 | 51.04 ± 1.11 | 51.88 ± 0.22 | 36.67 ± 0.35 | 64.10 ± 0.69 | 54.46 ± 1.49 |
| $\mathcal{I}_{\texttt{Arnoldi}}$ | 76.09 ± 0.66 | 93.76 ± 0.17 | 77.64 ± 0.93 | 62.20 ± 2.00 | 53.64 ± 0.67 | 38.8 ± 0.39 | 80.85 ± 1.52 | 64.62 ± 2.84 |
| $\mathcal{I}_{\texttt{GEX}}$ | **99.83 ± 0.03** | **99.83 ± 0.03** | **86.82 ± 1.24** | **81.98 ± 1.58** | **98.96 ± 0.01** | **97.69 ± 0.02** | **99.32 ± 0.19** | **97.51 ± 0.55** |

In this section, we apply the cross-influence on test dataset $S_{\text{test}}$ defined as

$$\mathcal{I}(z, S_{\text{test}}) := \frac{1}{|S_{\text{test}}|} \sum_{z' \in S_{\text{test}}} \mathcal{I}(z, z')$$

for the noisy label detection task in Appendix F.4. As the overfitting/memorization of mislabeled samples causes performance degradation for NNs, the estimated cross-influence values should be negative for these samples. Table 13 shows that IF approximations represent this tendency correctly. Mislabeled samples are best distinguished from clean samples using $\mathcal{I}_{\texttt{GEX}}$.

## F.6 TracInRP with Geometric Ensemble checkpoints

Table 14: Noisy label detection performance for $\mathcal{I}_{\texttt{TracInRP}}$ with Geometric Ensemble checkpoints

| Detection method | MNIST | | CIFAR-10 | | CIFAR-100 | | SVHN | |
|---|---|---|---|---|---|---|---|---|
| | AUC | AP | AUC | AP | AUC | AP | AUC | AP |
| $\mathcal{I}_{\texttt{RandProj}}$ | 70.90 ± 0.48 | 20.61 ± 0.46 | 62.70 ± 0.19 | 17.90 ± 0.17 | 79.96 ± 0.32 | 26.25 ± 0.47 | 58.82 ± 0.28 | 17.94 ± 0.19 |
| $\mathcal{I}_{\texttt{TracInRP}}$ | 94.37 ± 0.12 | 68.87 ± 0.97 | 89.56 ± 0.14 | 44.26 ± 0.37 | 74.99 ± 0.25 | 21.62 ± 0.26 | 91.52 ± 0.16 | 47.25 ± 0.77 |
| $\mathcal{I}_{\texttt{TracInRP}}$ with GE | 99.78 ± 0.03 | 97.00 ± 0.47 | 90.96 ± 0.14 | 41.52 ± 0.20 | 88.84 ± 0.22 | 48.91 ± 0.85 | 85.35 ± 0.20 | 23.06 ± 0.23 |
| $\mathcal{I}_{\texttt{GEX}}$ | 99.98 ± 0.00 | 99.87 ± 0.01 | 99.74 ± 0.02 | 98.31 ± 0.06 | 99.33 ± 0.03 | 96.08 ± 0.12 | 99.74 ± 0.00 | 96.59 ± 0.06 |

As discussed in Sec. 2, $\mathcal{I}_{\texttt{TracIn}}$ and $\mathcal{I}_{\texttt{TracInRP}}$ are not purely post-hoc as they require intermediate checkpoints during pre-training. Motivated by $\mathcal{I}_{\texttt{GEX}}$, one can propose to replace the intermediate checkpoints of $\mathcal{I}_{\texttt{TracInRP}}$ with post-hoc checkpoints generated by Geometric Ensemble. In this section, we evaluate this setting for the noisy label detection task in Sec. 5.1. Table 14 shows that this modification consistently outperforms $\mathcal{I}_{\texttt{RandProj}}$, which uses the last checkpoint $\theta^*$ only. Also, $\mathcal{I}_{\texttt{TracInRP}}$ with GE performs better than the original $\mathcal{I}_{\texttt{TracInRP}}$ except SVHN. However, this improvement does not outperform $\mathcal{I}_{\texttt{GEX}}$.

## F.7 Further ImageNet results

Table 15: Relabeled accuracy for ImageNet [12]

| | ViT-S-32 | Mixer-S-32 |
|---|---|---|
| Clean acc. | 67.83% | 64.37% |
| Noisy acc. | 63.42% | 61.84% |
| Relabeled with EL2N | 63.18% | 63.16% |
| Relabeled with GEX | **66.17%** | **63.45%** |

Table 16: Pruned acc. for ImageNet [12]

| | Mixer-S-32 |
|---|---|
| Full sample acc. | 67.83% |
| Pruned with EL2N | 54.87% |
| Pruned with GEX | **56.34%** |

Our first additional ImageNet experiment is the relabeling task presented in Sec. 5.2 on the ImageNet-1K environment with ViT and MLP-Mixer. To this end, we follow the relabeling process in Sec. 5.2 with the estimated influence in Table 2. As ViT and Mixer use multi-task binary classification loss instead of cross-entropy loss, the relabel function in (16) cannot be applied to this setting. To mitigate this issue, we use well-known distillation loss [21] for detected influential samples. Therefore, these influential samples receive a milder signal than from the hard label of the initial training. Table 15 presents the relabeled test accuracy for $\mathcal{I}_{\texttt{GEX}}$ and EL2N (the best method for noisy label detection except ours). It shows that $\mathcal{I}_{\texttt{GEX}}$ can detect mislabeled samples that require relabeling more accurately than EL2N. The second additional ImageNet experiment is the dataset pruning task in Sec. 5.3 on the ImageNet-1K with MLP-Mixer. For this purpose, we reproduce Mixer-S-32 and estimate the self-influence of $\mathcal{I}_{\texttt{GEX}}$ and EL2N score (which verified its scalability on the dataset pruning task in Sorscher et al. [58]). Then, we prune 512,466 samples (40%) with the smallest self-influence in ImageNet-1K and retrain neural networks with these pruned datasets. Table 16 presents the pruned test accuracy for $\mathcal{I}_{\texttt{GEX}}$ and EL2N. Similar to the results shown in Fig. 4, $\mathcal{I}_{\texttt{GEX}}$ demonstrates more effective identification of prunable samples than EL2N on the scalable ImageNet-1K dataset. Additionally, it is worth noting that EL2N cannot make use of open-source checkpoints and requires a computational cost of $(10 \sim 20$ epochs$) \times$ (number of checkpoints) from an initialized neural network. In summary, the better-pruned accuracy and the lower computational cost further illustrate the effectiveness of $\mathcal{I}_{\texttt{GEX}}$ in scalable settings.

# G    Limitations and Broader Impacts

While $\mathcal{I}_{\text{GEX}}$ effectively captures the multi-modal nature of $\mathcal{I}_{\text{LOO}}$'s self-influence distribution, we do not delve into the theoretical guarantees, such as approximation bounds, for $\mathcal{I}_{\text{GEX}}$. Nevertheless, we believe that this concern can be addressed in future research by making assumptions about the SGD steps used in generating checkpoints. Furthermore, although the data pruning results of $\mathcal{I}_{\text{GEX}}$ are significantly superior to other methods approximating IF, they do not outperform the performance achieved by state-of-the-art techniques like F-score and EL2N. This issue can be resolved by implementing the iterative pruning approach proposed in Harutyunyan et al. [19].

Our study examines the distributional bias present in current IF approximations and proposes an efficient solution. As a result, our approach has broad applicability in different scenarios. Specifically, our method offers enhanced efficiency in identifying mislabeled and prunable samples compared to existing approaches. Consequently, we believe that our method can contribute to enhancing and expediting the training process of energy-intensive deep learning models.

