# OpenReview forum: "GEX: A flexible method for approximating influence via Geometric Ensemble"
_NeurIPS.cc/2023/Conference — NeurIPS 2023 poster_

### Official Review · Reviewer_KU2o · 2023-07-07

**Soundness:** 3 good
**Presentation:** 3 good
**Contribution:** 3 good
**Rating:** 7
**Confidence:** 3

**Summary:**

This work studies approximations methods for data influence. The work identifies a common theoretical drawback behind popular approximation methods to Influence Function (IF) which suppresses their expressive power and affects their performance.

This work proposes a novel interpretation of existing IF approximations as some special form of Laplace approximation (LA) and points out that the drawback is due to the linearity of gradients and the singularity of Hessian. This work proposes an original IF approximation method that circumvents these issues, which removes the linearization to ease the bilinear constraint and leverages Geometric Ensemble (GE) for non-linear losses. Both conceptually and empirically, this work demonstrates significant improvement over existing IF approximation methods. The work includes a variety of experiments including many use cases.

**Strengths:**

The paper is nicely written. The narrative is smooth and the conceptual development is attractive. The problem being considered is of interest and finding of common drawback behind the mechanism of existing IF approximations is novel and insightful. Visualizations of conceptual findings are very helpful and give a straightforward presentation to help to understand.

Evaluations are thorough. Various use cases are considered and the empirical performance is satisfactory. Comprehensive details for experiments are given in Appendix.

**Weaknesses:**

Influence Function (IF) refers to the celebrated method proposed in < Pang Wei Koh and Percy Liang. Understanding black-box predictions via influence functions. In International conference on machine learning, pages 1885–1894. PMLR, 2017.> that is used to estimate the quantity in Eq. (1). Equation (1) is essentially the definition for LOO. It is NOT the "ground truth" of Influence Function (IF). Eq. (2) is the "Influence Function". It is NOT named "I_{Hess}".

The transition to Section 4.1 needs more elaborations. I_GEX proposed in this subsection is the core contribution of the paper, yet it isn't thoroughly discussed before moving on to discussing the limitation of other methods. Similarly, an introduction to LA is much needed as it lays important grounds for the development of this work. I may consider increasing my score if this can be reasonably addressed.

Though the conceptual flow of the paper is smooth, some important parts remain unclear. For example, what is Eq. 9 exactly and how is it calculated in actual implementation? How to conceptually interpret Eq. 10 and how it differs from Eq. 12?

**Questions:**

How does the proposed method compare to TracIn (not TracInRP)? And how does the result compare to the actual counterfactual (LOO)?

Appendix should not be submitted within the main paper, which has a page limit of 9.

What is the "retraining cost of IF"? IF does not need training or "retraining".

**Limitations:**

See Weaknesses.

---

> ### Author Rebuttal · Authors · 2023-08-09
>
> We appreciate your careful review and constructive comments! We give point-to-point replies to your comments in the following. Official comments will address questions that could not be answered due to the character limit.
>
> * Q1. [**Terminology**] Influence Function (IF) refers to the celebrated method proposed in < Pang Wei Koh and Percy Liang. Understanding black-box predictions via influence functions. In International conference on machine learning, pages 1885–1894. PMLR, 2017.> that is used to estimate the quantity in Eq. (1). Equation (1) is essentially the definition for LOO. It is NOT the "ground truth" of Influence Function (IF). Eq. (2) is the "Influence Function". It is NOT named "I_{Hess}".
>
> * A1. As the reviewer mentioned, [1] introduced Eq. (2) as an Influence Function (IF) to provide an approximation for the counterfactual effect of LOO retraining (Eq. (1)). In our work, we used the notation $ \mathcal{I}\_\mathtt{GT} $ and $ \mathcal{I}\_\mathtt{Hess} $ since the similar notations were originally introduced in [2] to emphasize that Eq. (2) approximates Eq. (1) using Hessian. However, we understand that this notation might be misleading for some readers. In response to the reviewer's feedback, we will revise the manuscript, denoting Eq. (1) as $\mathcal{I}\_\mathtt{LOO}$ and Eq. (2) as $\mathcal{I}$. Moreover, we believe that with this modification, it would become clear that the IF (Eq. (2)) does not require retraining, which addresses the following question: "What is the 'retraining cost of IF'? IF does not need training or 'retraining'."
>
>   [1] Koh, Pang Wei, and Percy Liang. "Understanding black-box predictions via influence functions." International conference on machine learning. PMLR, 2017.
>
>   [2] Schioppa, Andrea, et al. "Scaling up influence functions." Proceedings of the AAAI Conference on Artificial Intelligence. Vol. 36. No. 8. 2022.
>
> * Q2. [**Elaboration for Section 4.1**] The transition to Section 4.1 needs more elaborations. I_GEX proposed in this subsection is the core contribution of the paper, yet it isn't thoroughly discussed before moving on to discussing the limitation of other methods. Similarly, an introduction to LA is much needed as it lays important grounds for the development of this work. I may consider increasing my score if this can be reasonably addressed.
>
> * A2. Thank you for the helpful suggestions. To provide a clear motivation for GEX, we will revise the introduction of Section 4 (Lines 155-162) as follows:
>   To mitigate the distributional bias in Section 3, we propose a flexible IF approximation method using Geometric Ensemble (GE; [15]), named Geometric Ensemble for sample eXplanataion (GEX). Here is a summary of how GEX is developed.
>   $$
>   \mathcal{I}
>   \overset{\texttt{Delinearization}}{\underset{\texttt{Section 4.1}}{\longrightarrow}}
>   \mathcal{I}\_\mathtt{LA}
>   \overset{\texttt{LA to GE}}{\underset{\texttt{Section 4.2}}{\longrightarrow}}
>   \mathcal{I}\_\mathtt{GEX}
>   $$
>   In Section 4.1, we ensure that the influence approximation is not a bilinear form for the gradient by replacing gradients in IF with sample-loss deviations. The theoretical foundation for this step is provided by our Theorem 1 below, which establishes a relationship between the IF and the Laplace approximation (LA; [36]). Moving on to Section 4.2, we modify the parameter distribution to compute the sample-loss deviation from LA to GE. This modification is necessary because GE is based on the local geometry of the loss landscape around $\theta^*$, similar to LA while avoiding overestimating loss deviations caused by the singularity of the Hessian.
>
>   Also, the revised manuscript will include the following introduction to LA after Theorem 4.1 to assist readers:
>   The LA was proposed to approximate the posterior distribution with a Gaussian distribution. Recently, it has gained significant attention due to its simplicity and reliable calibration performance [1, 2]. Intuitively, LA is equivalent to the second-order Taylor approximation of log-posterior at $ \theta^* $ with Gaussian prior defined as $p(\psi):= \mathcal{N}(\psi| \mathbf{0}_P, \gamma^{-1} \mathbf{I}_P )$:
>   $$
>       \log p(\theta|S)
>       = \log p(S|\theta) + \log p(\theta) - \log Z \\
>       = - N\cdot L(S, \theta) + \log p(\theta) - \log Z \\
>       \approx - N\cdot L(S, \theta^*) - \frac{1}{2}(\theta-\theta^*)^{\top}(N\cdot H + \gamma \mathbf{I}_P )(\theta-\theta^*) - \log Z \\
>       = \log p(\theta^*|S) - \frac{1}{2}(\theta-\theta^*)^{\top}(N\cdot H + \gamma \mathbf{I}_P )(\theta-\theta^*). \\
>   $$
>   Here, the training loss represents the negative log-likelihood $L(S, \theta) = -\frac{1}{N}\log p(S| \theta)$, and $Z:= \int p(\theta) p(S|\theta) d\theta$ represents the evidence in Bayesian inference [3]. Similar to the IF, LA becomes computationally intractable when dealing with modern architectures due to the complexity of the Hessian matrix. To address this computational challenge, recent works have proposed various sub-curvature approximations, such as KFAC [1] and sub-network [2], which provide computationally efficient alternatives for working with LA.
>
>   [1] Ritter, Hippolyt, Aleksandar Botev, and David Barber. "A scalable laplace approximation for neural networks." 6th International Conference on Learning Representations, ICLR 2018-Conference Track Proceedings. Vol. 6. International Conference on Representation Learning, 2018.
>
>   [2] Erik Daxberger, Eric Nalisnick, James U Allingham, Javier Antoran, and Jose Miguel Hernandez- Lobato. Bayesian deep learning via subnetwork inference. In Marina Meila and Tong Zhang (eds.), Proceedings of the 38th International Conference on Machine Learning, volume 139 of Proceedings of Machine Learning Research, pp. 2510–2521. PMLR, 18–24 Jul 2021b.
>
>   [3] Bishop, Christopher M., and Nasser M. Nasrabadi. Pattern recognition and machine learning. Vol. 4. No. 4. New York: springer, 2006.

---

> > ### Author Response · Authors · 2023-08-10
> >
> > * Q3. [**Clarifications for Eq. (9) - (12)**] Though the conceptual flow of the paper is smooth, some important parts remain unclear. For example, what is Eq. 9 exactly and how is it calculated in actual implementation? How to conceptually interpret Eq. 10 and how it differs from Eq. 12?
> >
> > * A3. Eq. (9) in our paper is the empirical distribution of Geometric Ensemble (GE) [1]. To clarify further, we utilize the Dirac delta distribution to represent the empirical parameter distribution of GE. This is a common usage of the Dirac delta distribution, as mentioned in p.64 of [2]: "A common use of the Dirac delta distribution is as a component of an empirical distribution, ~.". To construct the empirical distribution $\\{ \theta^{m} \\}_{m=1}^{M}$, we collect intermediate checkpoints during the iterative SGD updates. For more details on the construction of GE and computing GEX based on GE, please refer to Appendix C.2.
> >
> >   Regarding Eq. (10), it can be intuitively understood as capturing the covariance of sample loss between two instances $z$ and $z'$. To ensure that the starting point used in calculating the covariance aligns with the mean value, we employ the concept of "sample loss deviation" instead of conventional covariance. This choice enables us to set the starting point as the sample loss at $\theta^*$. The main distinction between Eq. (10) and Eq. (12) lies in the parameter distribution used to calculate the expectation – LA or GE. This difference becomes crucial for dealing with non-linear sample loss deviation since LA tends to overestimate the sample loss deviation due to the singularity of the Hessian, as discussed in Section 4.2.
> >
> >   [1] Garipov, Timur, et al. "Loss surfaces, mode connectivity, and fast ensembling of dnns." Advances in neural information processing systems 31 (2018).
> >
> >   [2] Goodfellow, Ian, Yoshua Bengio, and Aaron Courville. Deep learning. MIT press, 2016.
> >
> > * Q4. [**Comparison to other methods**] How does the proposed method compare to TracIn (not TracInRP)? And how does the result compare to the actual counterfactual (LOO)?
> >
> > * A4. We refer to G3 [**Additional experiments for TracIn**] in the Global Rebuttal for these results.
> >
> > * Q5. [**Formatting**] Appendix should not be submitted within the main paper, which has a page limit of 9.
> >
> > * A5. Thank you for bringing up this concern. We acknowledged the problem right after the submission deadline and contacted the Program Chair about this issue. As per the Program Chair's response, NeurIPS 2023 will not reject papers with an appendix as long as it is evident that the main paper concludes on page 9. In line with the conference guidelines, we will separate the Appendix from the main text in the revised manuscript.

---

> > > ### Comment · Reviewer_KU2o · 2023-08-19
> > > **Thanks for the rebuttal**
> > >
> > > I have read through the reviews as well as the author's response. I appreciate the authors for their dedicated work and thanks for the response to my comments and for providing additional results.
> > >
> > > My questions have been adequately discussed and I have no further comments at this moment. I raised my score in support of this work.
> > >
> > > As for the notations for the influence approximations, I acknowledge the authors' point that they may have different references in different contexts for better distinctions. But I want to stress the importance of notation clarity as it could cause confusion to a more general audience at times.
> > >
> > > I hope the authors compile the new results and additional discussions into the paper or its Appendix and put in the effort to improve the presentation of the paper in light of the reviews.
> > >
> > >
> > > Nice work and good luck,
> > > Reviewer KU2o

---

> > > > ### Author Response · Authors · 2023-08-19
> > > >
> > > > Dear Reviewer KU2o,
> > > >
> > > > We greatly appreciate your valuable suggestions and positive evaluation of our paper! In response to your feedback, we will clarify the notation of influence following your comment ($\mathcal{I}$ for Influence Function and $\mathcal{I}\_\mathtt{LOO}$ for LOO counterfactual effect). Other discussions (e.g., experimental results) in our rebuttal will be properly included in the revised manuscript.
> > > >
> > > > Thank you again for your service to the community.
> > > >
> > > > Warm regards,
> > > >
> > > > The authors of submission 4926.

---

### Official Review · Reviewer_FeGS · 2023-07-07

**Soundness:** 3 good
**Presentation:** 3 good
**Contribution:** 3 good
**Rating:** 7
**Confidence:** 3

**Summary:**

The paper provide a novel connection between IF approximations and LA, and introduces a new IF approximation method. The motivation takes advantage from two observations, the removal of linearization can alleviate the bilinear constraint and, the utilization of Geometric Ensemble is advantageous.  Empirical results demonstrates the advantage of the proposed method with reduced computational complexity.

**Strengths:**

The paper provide a novel connection between IF approximations and LA, and introduces a new IF approximation method. The motivation takes advantage from two observations, the removal of linearization can alleviate the bilinear constraint and, the utilization of Geometric Ensemble is advantageous.  Empirical results demonstrates the advantage of the proposed method with reduced computational complexity.

I enjoyed a lot reading the literature review and the analysis offered by the paper. The paper also introduced several application scenarios where the proposed method demonstrated obvious performance boost, including the noisy label identification task.

**Weaknesses:**

W1. The  method is only justified on small dataset such as MIST and SVHN. Is it possible to verify the effectiveness of the method on larger dataset such as ImageNet1K?

**Questions:**

Please see weakness about the scale of the training data above.

**Limitations:**

Yes.

---

> ### Author Rebuttal · Authors · 2023-08-09
>
> We appreciate your valuable comments and suggestions! We provide a detailed reply to your questions in the following.
>
> * Q1. [**Scalability of GEX**] W1. The method is only justified on small dataset such as MIST and SVHN. Is it possible to verify the effectiveness of the method on larger dataset such as ImageNet1K?
>
> * A1. Following reviewer FeGS's recommendation, we verify the scalability of GEX for various tasks. In fact, Table 2 in our original submission already provides the evidence that the effectiveness of GEX in Table 1 scales well with larger datasets by showing that GEX outperforms the baselines in noisy label detection tasks on ImageNet-1K with ViT and MLP-Mixer. In the rebuttal phase, we further validate the scalability of GEX by extending the relabeling task (Section 5.1) and dataset pruning task (Section 5.2) to the ImageNet-1K setting.
>
>   Our first additional experiment is the relabeling task presented in Section 5.2 on the ImageNet-1K environment with ViT and MLP-Mixer. To this end, we follow the relabeling process in Section 5.2 with the estimated influence in Table 2. The following table presents the relabeled test accuracy for GEX and EL2N (the best method for noisy label detection except ours).
>   * **Table A. Relabeled accuracy for mislabeled samples**
>
>     |                     |       ViT-S-32      | MLP-Mixer-S-32 |
>     |:-------------------:|:-------------------:|:--------------:|
>     |      Clean acc.     |        67.83%       |     64.37%     |
>     |      Noisy acc.     |        63.42%       |     61.84%     |
>     | Relabeled with EL2N |        63.18%       |     63.16%     |
>     |  Relabeled with GEX |      **66.17%**     |   **63.45%**   |
>
>   Table A shows that GEX can detect mislabeled samples that require relabeling more accurately than EL2N.
>
>   The second additional experiment is the dataset pruning task in Section 5.4 on the ImageNet-1K with MLP-Mixer. For this purpose, we reproduce Mixer-S-32 and estimate the self-influence of GEX and EL2N score (which verified its scalability on the dataset pruning task in [1]). Then, we prune 512,466 samples (40%) with the smallest self-influence in ImageNet-1K and retrain neural networks with these pruned datasets. The following table presents the pruned test accuracy for GEX and EL2N:
>   * **Table B. Pruned accuracy for mislabeled samples**
>
>     |                  | MLP-Mixer-S-32 |
>     |:----------------:|:--------------:|
>     | Full sample acc. |     67.83%     |
>     | Pruned with EL2N |     54.87%     |
>     |  Pruned with GEX |   **56.34%**   |
>
>   Similar to the results shown in Figure 4, GEX demonstrates more effective identification of prunable samples than EL2N on the scalable ImageNet-1K dataset. Additionally, it is worth noting that EL2N cannot make use of open-source checkpoints and requires a computational cost of (10~20 epochs) x (number of checkpoints) from an initialized neural network. In summary, the better pruned accuracy and the lower computational cost further illustrate the effectiveness of GEX in scalable settings. We will include these results in the revised version.
>
>   [1] Sorscher, Ben, et al. "Beyond neural scaling laws: beating power law scaling via data pruning." Advances in Neural Information Processing Systems 35 (2022): 19523-19536.

---

> > ### Comment · Reviewer_FeGS · 2023-08-20
> > **Thanks for the reply.**
> >
> > Thanks for the reply! I will maintain my score.
> >
> > Best

---

> > > ### Author Response · Authors · 2023-08-21
> > >
> > > Dear Reviewer FeGS,
> > >
> > > We appreciate your confirmation of our rebuttal and the positive evaluation of our paper!
> > >
> > > Best regards,
> > >
> > > Authors of submission 4926.

---

### Official Review · Reviewer_TEqY · 2023-07-07

**Soundness:** 3 good
**Presentation:** 2 fair
**Contribution:** 3 good
**Rating:** 5
**Confidence:** 3

**Summary:**

This paper proposes a new method for approximating influential examples. It treats losses as non-linear functions, addresses the singularity problem of hessians and does not require multiple checkpoints or JVP computations for the influence. The authors also highlight a connection between Influence Functions (IF) and Laplace Approximation (LA) and discuss its limitations. They propose an approximation using Geometric Ensembles (GE) instead. The authors show empirically that their approach outperforms existing IF approximation methods on downstream tasks for noisy label detection, relabeling, dataset pruning and data source separation.

**Strengths:**

+ The paper aims to address two important limitations of influential examples: singular nature of hessians and linearity of influence functions.
+ The paper is well written, related work and existing limitations are properly discussed.
+ The paper also discusses and benchmarks their approach on several important downstream tasks including noisy label detection, relabeling and dataset pruning on several datasets and models.



**Weaknesses:**

+ The authors emphasize bilinear approximation but they do not explain what bilinearity applies to in the approximations. It would be good to explain it briefly to make it clear.
+ The overall motivation of GEX using dirac distribution in section 4.1 seems a bit unclear. It seems that GEX is highlighted by surfacing the connection between IF and Laplace Approximation (LA) and the limitations of p_{LA} but it is still unclear why GE distribution (Eq. 9) was chosen.
+ It’s a bit challenging to follow the descriptions of Figures 2 and 3 since the authors refer both to influence and self-influence. Self-influence also means potentially mislabeled and it is unclear whether it is what the authors mean. Typical vs influential annotations on those figures are also a bit unclear. What does typical mean in this case ?

**Minor comments**
+ Figure 2: Are 2a - 2d: axes annotations are missing.  Perhaps it would be good to mention that these are histograms in advance or in figure description.
+ Eq 9: parameter \psi is not explained in that section.


**Questions:**

+ Line 167: what’s the motivation of choosing Dirac delta distribution
+ Figure 2: Are 2a - 2d computed for self-influence or influence w.r.t. Test examples ? Have we compared I_GEX with I_Hess for influence on test examples ?  What does typical vs influential mean for self-influence ? High self-influence means that these are potentially mislabeled. So we want to say that the green ones are mislabeled ?
+ In terms of evaluation were there any experiments made for the original tracin without random projections for the Table 1.


**Limitations:**

The limitations of the work are not discussed.

---

> ### Author Rebuttal · Authors · 2023-08-09
>
> Thanks for reviewing our paper so thoroughly. We appreciate your feedback and would like to provide point-to-point replies to your questions in the following.
>
> * Q1. [**Bilinearity used in IF approximations**] The authors emphasize bilinear approximation but they do not explain what bilinearity applies to in the approximations. It would be good to explain it briefly to make it clear.
>
> * A1. The bilinearity in our paper refers to the influence approximations being bilinear with respect to the sample-wise gradients. Consequently, the self-influence based on these bilinear metrics (Eq. (2), (5), (6), and (7)) is quadratic for sample-wise gradients. Intuitively, bilinear approximation methods can be understood as inner products of sample-wise gradients with additional consideration of curvature. We will revise the manuscript to incorporate this explanation in Line 118.
>
> * Q2. [**Rationale for using GE**] The overall motivation of GEX using dirac distribution in section 4.1 seems a bit unclear. It seems that GEX is highlighted by surfacing the connection between IF and Laplace Approximation (LA) and the limitations of p_{LA} but it is still unclear why GE distribution (Eq. 9) was chosen.
>
> * A2. We use the GE distribution because GE is suitable for expressing the local geometry of the loss landscape around $ \theta^* $, similar to LA. However, unlike LA, GE does not overestimate loss deviations due to the singularity of Hessian. To clarify this, we will revise the introduction of Section 4 (Lines 155-162) as follows:
>   To mitigate the distributional bias in Section 3, we propose a flexible IF approximation method using Geometric Ensemble (GE; [15]), named Geometric Ensemble for sample eXplanataion (GEX). Here is a summary of how GEX is developed.
>   $$
>   \mathcal{I}\_\mathtt{Hess}
>   \overset{\texttt{Delinearization}}{\underset{\texttt{Section 4.1}}{\longrightarrow}}
>   \mathcal{I}\_\mathtt{LA}
>   \overset{\texttt{LA to GE}}{\underset{\texttt{Section 4.2}}{\longrightarrow}}
>   \mathcal{I}\_\mathtt{GEX}
>   $$
>   In Section 4.1, we ensure that the influence approximation is not a bilinear form for the gradient by replacing gradients in IF with sample-loss deviations. The theoretical foundation for this step is provided by our Theorem 1 below, which establishes a relationship between the IF and the Laplace approximation (LA; [36]). Moving on to Section 4.2, we modify the parameter distribution to compute the sample-loss deviation from LA to GE. This modification is necessary because GE is based on the local geometry of the loss landscape around $\theta^*$, similar to LA while avoiding overestimating loss deviations caused by the singularity of the Hessian.
>
> * Q3. [**Elaborations for Figures 2-3**] It’s a bit challenging to follow the descriptions of Figures 2 and 3 since the authors refer both to influence and self-influence. Self-influence also means potentially mislabeled and it is unclear whether it is what the authors mean. Typical vs influential annotations on those figures are also a bit unclear. What does typical mean in this case? // Figure 2: Are 2a - 2d: axes annotations are missing. Perhaps it would be good to mention that these are histograms in advance or in figure description. // Figure 2: Are 2a - 2d computed for self-influence or influence w.r.t. Test examples ? Have we compared I_GEX with I_Hess for influence on test examples ? What does typical vs influential mean for self-influence ? High self-influence means that these are potentially mislabeled. So we want to say that the green ones are mislabeled ?
>
> * A3. Figures 2-3 are all **histograms of self-influence**. In this setting, we do not measure influence w.r.t. test examples. Generally, "Typical samples" are those in which the presence or absence of individual samples has no significant impact on the decision boundary (i.e., low self-influence). In our setting, the typical samples are the two outer circle samples with relatively high density in Figure 2(a). Conversely, "Influential samples" refer to individual instances that substantially influence the decision boundary, resulting in high self-influence. Hence, influential samples correspond to the inner circle in Figure 2(a), demonstrating relatively low density. Note that mislabeled samples also greatly impact the decision boundary. We will modify Figure 2 (a) as G4. [**Modified Figure 2 (a)**] in the Global rebuttal. Also, We will clarify this information with axes annotations in the caption of Figure 2 of the revised manuscript.
>
> * Q4. [**Motivation of choosing Dirac delta**] Eq 9: parameter \psi is not explained in that section. Line 167: what’s the motivation of choosing Dirac delta distribution
>
> * A4. $\psi \in \mathbb{R}^{P}$ is an arbitrary vector in the parameter space to denote the Dirac delta distribution and Gaussian distribution. We will clarify this in Line 167 of the revised version. We used the Dirac delta distribution to represent the empirical parameter distribution of GE. This is a common usage of the Dirac delta distribution, as mentioned in p.64 of [1]: "A common use of the Dirac delta distribution is as a component of an empirical distribution, ~"
>
>   [1] Goodfellow, Ian, Yoshua Bengio, and Aaron Courville. Deep learning. MIT press, 2016.
>
> * Q5. [**Comparison to TracIn**] In terms of evaluation were there any experiments made for the original tracin without random projections for the Table 1.
>
> * A5. We refer to G3 [**Additional experiments for TracIn**] in the Global Rebuttal for these results.
>
> * Q6. [**Limitations**] The limitations of the work are not discussed.
>
> * A6. We discuss our method's limitations and broader impacts in Appendix G. We will clarify this information after the discussion in Section 4.3 (Practical advantages of GEX) in the revised manuscript.

---

> > ### Comment · Reviewer_TEqY · 2023-08-16
> > **Reply to Rebuttal by Authors**
> >
> > Thank you very much for the detailed explanation. I increased the score by 1 point.

---

> > > ### Author Response · Authors · 2023-08-16
> > >
> > > Dear Reviewer TEqY,
> > >
> > > Thank you very much for your constructive suggestions and positive evaluation of the significance of our paper! Your suggestion will be included to enhance the readability of our paper for readers without sufficient background knowledge.
> > >
> > > Best regards,
> > >
> > > The Authors of Paper 4926

---

### Official Review · Reviewer_sNxP · 2023-07-07

**Soundness:** 3 good
**Presentation:** 2 fair
**Contribution:** 3 good
**Rating:** 6
**Confidence:** 2

**Summary:**

The paper identifies that existing influence functions suffer from a fundamental drawback due to their bilinear form. To address, this they propose an influence calculation that is nonlinear                                                    .

**Strengths:**

The GEX influences seem to outperform all baselines when detecting label noise on CIFAR10-like datasets.

The approach is motivated by principled failures of existing influence functions due to factors such as singularity of the Gaussian.

**Weaknesses:**

There is a lot of information pushed to the appendix that is reference in the main text, to the extent that it more difficult to really understand without having the appendix at hand. The current main text was too sparse for me to really understand any particular component.

It seems the experiments largely focused on MNIST/SVHN/CIFAR10 variants. It would be nice to see some kind of breadth in application or scalability to more than CIFAR10.

**Questions:**

I did not find it very clear what benefit or cards the geometric ensemble brings to the table.

It would be nice to have an actual example using only open source checkpoints, since this is one of the purported benefits of the research.

In the limitations, the authors believe their method can expedite the training process of energy-efficient deep learning models.

**Limitations:**

In the appendix.

---

> ### Author Rebuttal · Authors · 2023-08-09
>
> Thank you for your detailed review of our paper. We appreciate your feedback and would like to provide point-to-point replies to your questions in the following.
>
> * Q1. [**Lack of details in main text**] There is a lot of information pushed to the appendix that is reference in the main text, to the extent that it more difficult to really understand without having the appendix at hand. The current main text was too sparse for me to really understand any particular component.
>
> * A1. We apologize for not including all the necessary content within the main text due to the length constraints. We understand that relying heavily on the appendix might affect the readability. Our appendix contains additional material, such as proofs for propositions discussed in the main text, complexity analysis, experimental setups, and additional experimental results. We use this supplementary content to support the key points covered in the main paper, including the problem we address (distributional bias in Section 3) and the rationale behind our approach (Section 4). If you have any suggestions regarding specific content from the appendix that should be moved to the main text, please share them with us. We will take your feedback into careful consideration while preparing the revised version.
>
> * Q2. [**Scalability of GEX**] It seems the experiments largely focused on MNIST/SVHN/CIFAR10 variants. It would be nice to see some kind of breadth in application or scalability to more than CIFAR10.
>
> * A2. Following reviewer sNxP's recommendation, we verify the scalability of GEX for various tasks. In fact, Table 2 in our paper provides evidence that the effectiveness of GEX in Table 1 scales well with larger datasets by showing that GEX outperforms the baselines in noisy label detection tasks on ImageNet-1K with ViT and MLP-Mixer. In the rebuttal phase, we further validate the scalability of GEX by extending the relabeling task to the ImageNet-1K setting. We refer to G1 [**Additional experiments: Relabeling**] in the Global Rebuttal for these results.
>
> * Q3. [**Rationale for using GE**] I did not find it very clear what benefit or cards the geometric ensemble brings to the table.
>
> * A3. We use the GE distribution because GE is suitable for expressing the local geometry of the loss landscape around $ \theta^* $, similar to LA. However, unlike LA, GE does not overestimate loss deviations due to the singularity of Hessian. To clarify this, we will revise the introduction of Section 4 (Lines 155-162) as follows:
>   To mitigate the distributional bias in Section 3, we propose a flexible IF approximation method using Geometric Ensemble (GE; [15]), named Geometric Ensemble for sample eXplanataion (GEX). Here is a summary of how GEX is developed.
>   $$
>   \mathcal{I}\_\mathtt{Hess}
>   \overset{\texttt{Delinearization}}{\underset{\texttt{Section 4.1}}{\longrightarrow}}
>   \mathcal{I}\_\mathtt{LA}
>   \overset{\texttt{LA to GE}}{\underset{\texttt{Section 4.2}}{\longrightarrow}}
>   \mathcal{I}\_\mathtt{GEX}
>   $$
>   In Section 4.1, we ensure that the influence approximation is not a bilinear form for the gradient by replacing gradients in IF with sample-loss deviations. The theoretical foundation for this step is provided by our Theorem 1 below, which establishes a relationship between the IF and the Laplace approximation (LA; [36]). Moving on to Section 4.2, we modify the parameter distribution to compute the sample-loss deviation from LA to GE. This modification is necessary because GE is based on the local geometry of the loss landscape around $\theta^*$, similar to LA while avoiding overestimating loss deviations caused by the singularity of the Hessian.
>
> * Q4. [**Experiments using open-source checkpoints**] It would be nice to have an actual example using only open source checkpoints, since this is one of the purported benefits of the research.
>
> * A4. Following the recommendation of reviewer sNxP, we tried to conduct additional experiments by extending the dataset pruning task in Section 5.4 to ImageNet-1K with MLP-Mixer. However, as Mixer-B-16 and bigger models, which is beyond our computational budget, are released in the official repository, we decided to pre-train the smaller Mixer-S-32 model and conduct experiments. The reproduced accuracy of Mixer-S-32 is 64.37%, which is higher than the published accuracy of 63.9% [1, 2]. We refer to G2 [**Additional experiments: Dataset pruning**] in the Global Rebuttal for these results.

---

> > ### Comment · Reviewer_sNxP · 2023-08-14
> >
> > Thank you for the response! I appreciated the explanations and additional experiments scaling the approach and motivating the setting of open-source checkpoints.  As both the questions and weaknesses were addressed, I have updated my score accordingly.
> >
> > To help the authors clarify the main text, it is not strictly necessary that you bring appendix material to the main paper, instead just properly explain the referenced material. The proof and pseudocode references were fine, but I found the other references to the appendix that state "we do X in the appendix" with no motivation/reason to be largely unhelpful. For example, on L206 the cross entropy reference is just presented with no reason for it. The ablation referenced in L236, analysis referenced in L247, and cross influence referenced in L266 are just stated with no takeaway. The reference on L279 is at least properly motivated but again has no takeaway. These are just stated as is and come across as having no rhyme or reason.

---

> > > ### Author Response · Authors · 2023-08-14
> > >
> > > Dear Reviewer sNxP,
> > >
> > > We are grateful for your comprehensive suggestions and your positive evaluation of the significance of our paper! In response to your feedback, we will clarify the rationale behind referencing the Appendix in our revised manuscript. We believe this clarification will greatly assist readers in resolving any queries that may arise while reading the paper.
> > >
> > > Warm regards,
> > >
> > > The Authors of Paper 4926

---

### Official Review · Reviewer_8KEq · 2023-07-21

**Soundness:** 3 good
**Presentation:** 3 good
**Contribution:** 3 good
**Rating:** 6
**Confidence:** 2

**Summary:**

In this paper, the authors point out that standard approximations of Influence Function (IF) suffer from performance degradation due to oversimplified influence distributions caused by their bilinear approximation, suppressing the expressive power of samples with a relatively strong influence. Therefore, they propose a new interpretation of existing IF approximations as an average relationship between two linearized losses over parameters sampled from the Laplace approximation (LA). By doing so, they highlight two limitaions and propose GEX to improve them. The proposed GEX outperforms standard IF approximations in downstream tasks, including noisy label detection, relabeling, dataset pruning, and data source separation.


**Strengths:**

1. This paper is well written. The details of movtivation, methodology, experiments and proofs are described clearly.
2. The insight of this paper is quite reasonable and the proposed approach looks novel.
3. The proposed method is verified on a wide range of downstream tasks, including noisy label detection, relabeling, dataset pruning, and data source separation.

**Weaknesses:**

1. In table 2, the gap between GEX and the other settings is very large. It seems that the other settings are not specificly designed for this task. Is this comparison meaningful?
2. Except table 2, the other experiments are executed on some small and relatively easy dataset, I am not sure whether the conclusions from these results are solid enough.


**Questions:**

1. I think nosiy label detection combined with relabeling mislabeled samples is very useful in many practical scenarios. But the experiments are only executed on CIFAR, which is a quite small dataset. What is the result if we use GEX to detection noisy labels in Imagenet and relabel them?
2. How can this approach help some pratical scenarios?

**Limitations:**

See weakness. Most experiments are executed on small and relatively easy datasets. The effectiveness of this approach on those more challenging and large benchmarks is not explored.

---

> ### Author Rebuttal · Authors · 2023-08-09
>
> We appreciate your careful review and constructive comments! We give point-to-point replies to your comments in the following.
>
> * Q1. [**Scalability issue of other IF approximations**] In table 2, the gap between GEX and the other settings is very large. It seems that the other settings are not specificly designed for this task. Is this comparison meaningful?
>
> * A1. The Influence Function (IF), EL2N, and F-score are known to be applicable for noisy label detection [1, 2, 3], which can be confirmed by Table 1 of our paper. Table 2 aims to investigate whether the findings from Table 1 are scalable to large-scale datasets. This table shows that other influence approximations exhibit significant performance degradation compared to GEX. A similar scalability issue was pointed out in [4] in the context of dataset pruning. The scalability issue also exists for non-influence approximation methods: F-score fails in cases where certain samples were never correctly predicted during pre-training, as mentioned in Lines 283-284. Since EL2N is based on the Brier score (i.e., L2 distance of prediction and label) at the early training phase, there is a performance degradation in the ImageNet setting where the training speed is slow. However, GEX does not suffer from such a scalability issue.
>
>   [1] Koh, Pang Wei, and Percy Liang. "Understanding black-box predictions via influence functions." International conference on machine learning. PMLR, 2017.
>
>   [2] Mansheej Paul, Surya Ganguli, and Gintare Karolina Dziugaite. Deep learning on a data diet: Finding important examples early in training. Advances in Neural Information Processing Systems, 34:20596–20607, 2021
>
>   [3] Mariya Toneva, Alessandro Sordoni, Remi Tachet des Combes, Adam Trischler, Yoshua Bengio, and Geoffrey J Gordon. An empirical study of example forgetting during deep neural network learning. arXiv preprint arXiv:1812.05159, 2018.
>
>   [4] Sorscher, Ben, et al. "Beyond neural scaling laws: beating power law scaling via data pruning." Advances in Neural Information Processing Systems 35 (2022): 19523-19536.
>
> * Q2. [**Scalability of GEX**] Except table 2, the other experiments are executed on some small and relatively easy dataset, I am not sure whether the conclusions from these results are solid enough. // I think nosiy label detection combined with relabeling mislabeled samples is very useful in many practical scenarios. But the experiments are only executed on CIFAR, which is a quite small dataset. What is the result if we use GEX to detection noisy labels in Imagenet and relabel them?
>
> * A2. Following the recommendation of reviewer 8KEq, we further validate the scalability of GEX by extending the relabeling task (Section 5.1) and dataset pruning task (Section 5.2) to the ImageNet-1K setting. We refer to G1 [**Additional experiments: Relabeling**] and G2 [**Additional experiments: Dataset pruning**] in the Global Rebuttal for these results.
>
> * Q3. [**Practical scenarios for GEX**] How can this approach help some pratical scenarios?
>
> * A3. The tasks of noisy label detection, relabeling, and dataset pruning presented in Section 5 have several practical applications. One of the applications of GEX is to prune samples with minimal impact on training, allowing for more efficient training than conventional neural scaling laws, as described in [1]. Also, GEX can effectively detect and correct mislabeled samples in datasets obtained from (noisy) crowd annotations, where various people contribute to labeling (as demonstrated in Section 5.1-5.2).  For instance, recent research in the medical domain, particularly on chest X-rays [2, 3], shows that annotations made by doctors in public datasets are sometimes incorrect [2]. Thus, models trained with these datasets can be improved by retraining or fine-tuning with corrected labels [3]. Considering that GEX improves performance in a general relabeling benchmark, it is reasonable to expect that it will also enhance performance in medical applications.
>
>   [1] Sorscher, Ben, et al. "Beyond neural scaling laws: beating power law scaling via data pruning." Advances in Neural Information Processing Systems 35 (2022): 19523-19536.
>
>   [2] Tang, Siyi, et al. "Data valuation for medical imaging using Shapley value and application to a large-scale chest X-ray dataset." Scientific reports 11.1 (2021): 8366.
>
>   [3] Kim, Doyun, et al. "Accurate auto-labeling of chest X-ray images based on quantitative similarity to an explainable AI model." Nature communications 13.1 (2022): 1867.
>
>   [4] Wang, Xiaosong, et al. "Chestx-ray8: Hospital-scale chest x-ray database and benchmarks on weakly-supervised classification and localization of common thorax diseases." Proceedings of the IEEE conference on computer vision and pattern recognition. 2017.

---

> > ### Comment · Reviewer_8KEq · 2023-08-15
> > **Reply to Authors**
> >
> > Thanks for your rebuttal. My concerns have been addressed and I would like to raise my rating to 6.

---

> > > ### Author Response · Authors · 2023-08-15
> > >
> > > Dear Reviewer 8KEq,
> > >
> > > We truly appreciate your suggestions and raising the score of our paper!
> > >
> > > Best regards,
> > >
> > > Authors of paper 4926

---

### Author Rebuttal · Authors · 2023-08-09

* G1. [**Additional experiments: Relabeling**] Our first additional experiment is the relabeling task presented in Section 5.2 on the ImageNet-1K environment with ViT and MLP-Mixer. To this end, we follow the relabeling process in Section 5.2 with the estimated influence in Table 2. The following table presents the relabeled test accuracy for GEX and EL2N (the best method for noisy label detection except ours).
  * **Table A. Relabeled accuracy for mislabeled samples**

    |                     |       ViT-S-32      | MLP-Mixer-S-32 |
    |:-------------------:|:-------------------:|:--------------:|
    |      Clean acc.     |        67.83%       |     64.37%     |
    |      Noisy acc.     |        63.42%       |     61.84%     |
    | Relabeled with EL2N |        63.18%       |     63.16%     |
    |  Relabeled with GEX |      **66.17%**     |   **63.45%**   |

  Table A shows that GEX can detect mislabeled samples that require relabeling more accurately than EL2N.

* G2. [**Additional experiments: Dataset pruning**]   The second additional experiment is the dataset pruning task in Section 5.4 on the ImageNet-1K with MLP-Mixer. For this purpose, we reproduce Mixer-S-32 and estimate the self-influence of GEX and EL2N score (which verified its scalability on the dataset pruning task in [1]). Then, we prune 512,466 samples (40%) with the smallest self-influence in ImageNet-1K and retrain neural networks with these pruned datasets. The following table presents the pruned test accuracy for GEX and EL2N:
  * **Table B. Pruned accuracy for mislabeled samples**

    |                  | MLP-Mixer-S-32 |
    |:----------------:|:--------------:|
    | Full sample acc. |     67.83%     |
    | Pruned with EL2N |     54.87%     |
    |  Pruned with GEX |   **56.34%**   |

  Similar to the results shown in Figure 4, GEX demonstrates more effective identification of prunable samples than EL2N on the scalable ImageNet-1K dataset. Additionally, it is worth noting that EL2N cannot make use of open-source checkpoints and requires a computational cost of (10~20 epochs) x (number of checkpoints) from an initialized neural network. In summary, the better pruned accuracy and the lower computational cost further illustrate the effectiveness of GEX in scalable settings. We will include these results in the revised version.

  [1] Sorscher, Ben, et al. "Beyond neural scaling laws: beating power law scaling via data pruning." Advances in Neural Information Processing Systems 35 (2022): 19523-19536.

* G3. [**Additional experiments for TracIn**] The downstream task performance of TracIn and TracInRP is similar, as mentioned in the TracIn paper [1]. In the rebuttal phase, we additionally measure the performance of TracIn on the noisy label detection (Section 5.1) and relabeling (Section 5.2) tasks to verify this. Due to the high time complexity associated with sample-wise gradients, we do not repeatedly measure the self-influence of TracIn. The following tables present the noisy label detection performance and relabeled accuracy achieved by TracIn with baselines (TracInRP and GEX).

  * AUC (Area Under the Curve) and AP (Average Precision) results for different datasets
  *
    |     AUC / AP    |     CIFAR-10 (synthetic)    |    CIFAR-100 (synthetic)    |          CIFAR-10-N         |         CIFAR-100-N         |
    |:---------------:|:---------------------------:|:---------------------------:|:---------------------------:|:---------------------------:|
    |      TracIn     |        89.98 / 43.21        |        75.53 / 22.25        |        76.48 / 64.69        |        68.91 / 55.86        |
    |     TracInRP    | 89.56 ± 0.14 / 44.26 ± 0.37 | 74.99 ± 0.25 / 21.62 ± 0.26 | 77.24 ± 0.45 / 65.17 ± 0.68 | 69.04 ± 0.28 / 56.41 ± 0.31 |
    |       GEX       | 99.74 ± 0.02 / 98.31 ± 0.06 | 99.33 ± 0.03 / 96.08 ± 0.12 | 96.20 ± 0.03 / 94.89 ± 0.04 | 89.76 ± 0.01 / 86.30 ± 0.01 |

  * Relabeled accuracy results for different datasets

    | Relabeled acc. | CIFAR-10 (synthetic) | CIFAR-100 (synthetic) |  CIFAR-10-N  |  CIFAR-100-N |
    |:--------------:|:--------------------:|:---------------------:|:------------:|:------------:|
    |     TracIn     |         91.24        |         72.07         |     68.36    |     54.87    |
    |    TracInRP    |     90.82 ± 0.06     |      71.70 ± 0.15     | 68.12 ± 0.23 | 55.20 ± 0.06 |
    |       GEX      |     93.54 ± 0.05     |      75.04 ± 0.10     | 73.94 ± 0.24 | 57.13 ± 0.10 |

  In accordance with Figure 1 in [1], we found that neither TracIn nor TracInRP consistently outperforms the other.

  The LOO counterfactual effect requires significant computational resources due to the necessity of LOO retraining for each sample. Consequently, we report the result of the LOO counterfactual only for the toy dataset in Figures 2-3. This computational challenge has been highlighted in various papers on Influence Functions [1, 2, 3]. To the best of our knowledge, no previous work used the LOO counterfactual effect for downstream tasks in Section 5.

  [1] Pruthi, Garima, et al. "Estimating training data influence by tracing gradient descent." Advances in Neural Information Processing Systems 33 (2020): 19920-19930.

* G4. [**Modified Figure 2 (a)**] We also provide a modified Figure 2 (a) to clarify the definition of "Typical" and "Influential" in Figures 2-3.

---

### Decision · Program_Chairs · 2023-09-21

**Decision:**

Accept (poster)

**Comment:**

This paper proposed a new algorithm for influence function approximation, aimed at addressing the issues of the previous approximations. It is demonstrated that the new algorithm outperformed previous methods on downstream tasks such as noisy label detection and dataset pruning. The analysis and formulation are convincing, and with additional results on larger datasets added during the rebuttal, the proposed method could be a promising alternative IF approximation with better performance and lighter computation.